# A Local Polyak-Łojasiewicz and Descent Lemma of Gradient Descent For Overparametrized Linear Models

**Ziqing Xu**                                            *ziqingxu@wharton.upenn.edu*
*Center for Innovation in Data Engineering and Science (IDEAS)*
*Department of Statistics and Data Science, the Wharton School*
*University of Pennsylvania*

**Hancheng Min**                                         *hanchmin@seas.upenn.edu*
*Center for Innovation in Data Engineering and Science (IDEAS)*
*University of Pennsylvania*

**Salma Tarmoun**                                        *starmoun@sas.upenn.edu*
*Center for Innovation in Data Engineering and Science (IDEAS)*
*Graduate Group in Applied Mathematics and Computational Science*
*University of Pennsylvania*

**Enrique Mallada**                                      *mallada@jhu.edu*
*Mathematical Institute for Data Science (MINDS)*
*Department of Electrical and Computer Engineering*
*Johns Hopkins University*

**René Vidal**                                           *vidalr@upenn.edu*
*Center for Innovation in Data Engineering and Science (IDEAS)*
*Department of Electrical and Systems Engineering; Department of Radiology*
*University of Pennsylvania*

**Reviewed on OpenReview:** *https://openreview.net/forum?id=VPl3T43Hxb*

## Abstract

Most prior work on the convergence of gradient descent (GD) for overparameterized neural networks relies on strong assumptions on the step size (infinitesimal), the hidden-layer width (infinite), or the initialization (large, spectral, balanced). Recent efforts to relax these assumptions focus on two-layer linear networks trained with the squared loss. In this work, we derive a linear convergence rate for training two-layer linear neural networks with GD for general losses and under relaxed assumptions on the step size, width, and initialization. A key challenge in deriving this result is that classical ingredients for deriving convergence rates for nonconvex problems, such as the Polyak-Łojasiewicz (PL) condition and Descent Lemma, do not hold globally for overparameterized neural networks. Here, we prove that these two conditions hold locally with local constants that depend on the weights. Then, we provide bounds on these local constants, which depend on the initialization of the weights, the current loss, and the global PL and smoothness constants of the non-overparameterized model. Based on these bounds, we derive a linear convergence rate for GD. Our convergence analysis not only improves upon prior results but also suggests a better choice for the step size, as verified through our numerical experiments.

## 1 Introduction

Neural networks have shown great empirical success in many real-world applications, such as computer vision He et al. (2016) and natural language processing (Vaswani et al., 2018). However, our theoretical understanding

of why neural networks work so well is still scarce. One unsolved question is why neural networks trained via vanilla gradient descent (GD) enjoy fast convergence although their loss landscape is non-convex. This question has motivated recent work which focuses on deriving convergence rates for overparameterized neural networks. However, most prior work on the linear convergence of GD for overparametrized neural networks requires strong assumptions on the step size (infinitesimal), width (infinitely large), initialization (large, spectral), or restrictive choices of the loss function (squared loss) (See Table 1 for details).

**Derivation of convergence of nonlinear networks requires restrictive assumptions.** The work of (Du et al., 2018b; Lee et al., 2019; Liu et al., 2022) studies the convergence of GD for neural networks in the neural tangent kernel (NTK) regime, which requires the network to have large or infinite width and large initialization. However, Chizat et al. (2019); Chen et al. (2022) show that the NTK regime limits feature learning, and the generalization performance of neural networks in this regime degrades substantially. To go beyond the NTK regime, Mei et al. (2018); Chizat & Bach (2018); Sirignano & Spiliopoulos (2020); Ding et al. (2022) study convergence of neural networks in the mean-field regime under the assumption of infinite width and infinitesimal step sizes. However, while such analysis can guarantee convergence to a global optimum for a wider range of initializations, it still imposes strong assumptions on the network width (infinite) and step size (infinitesimal).

**Suboptimal convergence rates of overparametrized linear networks.** To relax the assumptions on the width, step size, or initialization, some recent work focused on deriving convergence rates of GD for neural networks with a linear activation function in the context of matrix factorization (Arora et al., 2018; Du et al., 2018b) and linear regression (Du & Hu, 2019; Xu et al., 2023). In these settings, instead of learning a matrix $W$ directly, one learns an overparametrized version of $W$ defined as the product of $L$ matrices $W_1 \cdots W_L$. For example, in the case $L = 2$, which is the one we will analyze in this paper, this leads to the following non-overparametrized and overparametrized problems defined, respectively, by

$$\min_W \ell_{\text{LS}}(W) := \frac{1}{2}\|Y - XW\|_F^2, \qquad \text{(non-overparametrized)}$$

and

$$\min_{W_1, W_2} L_{\text{LS}}(W_1, W_2) := \frac{1}{2}\|Y - XW_1W_2\|_F^2, \qquad \text{(overparametrized)}$$

where $X, Y$ are data matrices, $W, W_1, W_2$ are weight matrices, and LS stands for least-square objective. Here, *overparametrized* indicates that, although the function classes represented by $XW$ and $XW_1W_2$ coincide when the sizes of $W_1$ and $W_2$ are large enough, $XW_1W_2$ introduces additional matrices to represent the same function. Therefore, while the non-overparametrized loss above defines a standard linear regression problem, the overparametrized loss can be seen as using a two-layer linear network to solve the regression problem.

A classical approach to derive a linear convergence rate for non-convex problems relies on the PL condition[1] and the smoothness inequality (see §2.1 for details). While these two conditions hold globally for non-overparametrized models, they do not hold globally for overparametrized linear networks (see §2.2 for details). This is because overparametrization skews the gradient through a weight-dependent linear operator $\mathcal{T}$ that acts on $\nabla \ell_{\text{LS}}$ (derived based on chain rule)

$$\nabla L_{\text{LS}}(W_1, W_2) = \nabla_{W_1, W_2} W \circ \nabla \ell_{\text{LS}} := \mathcal{T}(\nabla \ell_{\text{LS}}), \quad W = W_1 W_2. \tag{1}$$

To circumvent this challenge, existing studies (Arora et al., 2018; Du & Hu, 2019; Xu et al., 2023) use the smoothness inequality of the non-overparametrized model, which holds globally, as a substitute for the one of the overparametrized model. In addition, prior works use different methods and assumptions to derive (local) PL conditions (See Appendix C for details). For example, Arora et al. (2018) impose unrealistic assumptions on the initialization (large *margin*[2] and small *imbalance*[3]), while Du & Hu (2019) assume large initialization and large width. Xu et al. (2023) show the overparametrized models satisfy local PL conditions and control

---

[1]PL condition has been widely used to derive convergence for non-convex problems (Karimi et al., 2020; Arora et al., 2018; Min et al., 2021; Fridovich-Keil et al., 2023; Xu et al., 2023).

[2]The *margin* is a quantity that measure how close the initialization is to the global minimum.

[3]The *imbalance* is a quantity that measures the difference between the weights of two adjacent layers.

Table 1: Comparison with prior work

|  | Work | Loss | Step Size | Width | Initialization |
|---|---|---|---|---|---|
| Nonlinear networks | (Du et al., 2018b; Lee et al., 2019; Jacot et al., 2018; Liu et al., 2022; Nguyen & Mondelli, 2020) | Squared loss | **Finite** | Large | Large |
|  | (Mei et al., 2018; Chizat & Bach, 2018; Sirignano & Spiliopoulos, 2020; Ding et al., 2022) | Squared loss | Infinitesimal | Large | **General** |
| Linear networks | (Saxe et al., 2013; Gidel et al., 2019; Tarmoun et al., 2021) | Squared loss | Infinitesimal | **Finite** | Spectral |
|  | (Arora et al., 2018; Du et al., 2018a) | Squared loss | **Finite** | **Finite** | Large margin and small imbalance |
|  | (Xu et al., 2023) | Squared loss | **Finite** | **Finite** | **General** |
|  | This work | **General** | **Finite** | **Finite** | **General** |

these constants using the weights at initialization through careful choices of the step sizes. However, how the initialization scale and width of the network affect these constants is missing.

While the aforementioned analyses successfully derive linear convergence rates for overparametrized linear networks, all of them only consider squared loss and do not generalize to other types of loss functions. Moreover, the analysis techniques rely on the smoothness inequality of the non-overparametrized model. Therefore, they lack insight into the optimization geometry of overparameterized problems. Numerically, the actual convergence rate of the loss under the step sizes proposed in all the above work is slow (See §4.2 for details). Hence, an analysis is needed that establishes the convergence of neural networks trained using GD with a general loss under more relaxed assumptions. Furthermore, this analysis should offer more accurate predictions of the actual rates of convergence. This paper aims to bridge some of these gaps.

## 1.1 Main Contribution

In this work, we derive linear convergence rates for GD with possibly adaptive step sizes on overparameterized two-layer linear networks with a general loss, finite width, finite step size, and general initialization. Specifically, we make the following contributions:

- We analyze the Hessian of two-layer linear networks and show that the optimization problem satisfies a local PL condition and local Descent Lemma, where we characterize the local PL constant and local smoothness constant along the descent direction at GD iterates[4] by their corresponding loss values and the singular values of the weight matrices (See Theorem 3.1).

- We show that when the step size satisfies certain constraints (not infinitesimal), the *imbalance* remains close to its initial value. Based on this property, we prove the local PL and smoothness constants can be bounded along the trajectory of GD (See Theorem 3.2). Building on these results, we design an adaptive step size scheduler that yields a linear convergence rate for GD. Moreover, our results cover GD with decreasing, constant, and increasing step sizes while prior work (Arora et al., 2018; Du & Hu, 2019; Xu et al., 2023) only covers GD with decreasing and constant step sizes.

- We show that, under our step size scheduler, the local smoothness constant decreases along the GD trajectory, indicating that the optimization landscape becomes more benign as training proceeds. Building on this observation, we demonstrate that our step size scheduler accelerates convergence (see §4.2).

- Our analysis allows us to show that when GD iterates are around a global minimum, the difference between the local rate of convergence of the overparametrized model and the rate of the non-overparametrized model is up to one factor of the condition number of $\mathcal{T}_0$ (see §2.2 for definitions) which can be made arbitrarily

---

[4]In the paper, we adopt the term *local smoothness constant* as a convenient shorthand to refer to the smoothness constant along the descent direction at GD iterates.

close to one by proper initialization. In contrast, the (asymptotic) convergence rates derived in prior work (Arora et al., 2018; Du & Hu, 2019; Xu et al., 2023) are adversely affected by the square of the condition number of $\mathcal{T}_0$, leading to slower rates compared to our results.

## 1.2 Related work

We now provide a detailed description of prior work in addition to the discussion above.

One line of work (Du et al., 2018b; Lee et al., 2019; Liu et al., 2022; Nguyen & Mondelli, 2020) studies the convergence of GD with constant step size for squared loss under the assumption that the width and initialization of neural networks are sufficiently large, which is also known as the neural tangent kernel (NTK) regime. Under these assumptions, the training trajectories of a neural network are governed by a kernel determined at initialization and the network weights stay close to their initial values. Such properties help them derive a linear convergence rate of GD. However, the convergence rates derived in (Du et al., 2018b; Lee et al., 2019; Liu et al., 2022; Nguegnang et al., 2021) are inversely proportional to the number of samples or the initial loss and can be arbitrarily close to one when the number of samples or the initial loss is sufficiently large. Moreover, Chizat et al. (2019); Chen et al. (2022) show that the NTK regime prohibits feature learning, and the performance of neural networks in this regime degrades substantially.

To relax assumptions on width, step size, and initialization, numerous studies have explored the convergence rates of gradient-based algorithms for neural networks with linear activations. These studies are motivated by observations that linear networks exhibit similar nonlinear learning phenomena to those seen in simulations of nonlinear networks (Saxe et al., 2013). For instance, Du & Hu (2019); Arora et al. (2018); Xu et al. (2023) establish linear convergence rates for linear networks with squared loss optimized via GD. Their analyses rely on the smoothness inequality of non-overparametrized models and do not provide an exact characterization of the optimization landscape in overparametrized settings. In contrast, we deliver a tighter analysis by characterizing the local PL condition and Descent Lemma in overparametrized models, allowing us to derive faster convergence rates and design adaptive step sizes that adjust to the evolving local optimization landscape along the GD trajectory (see Appendix C for details). Moreover, Arora et al. (2018) and Du & Hu (2019) examine GD with constant step sizes; however, Arora et al. (2018) requires unrealistic initialization conditions (large margin and low imbalance), while Du & Hu (2019) assumes large initialization and width. These restrictive assumptions lead to convergence rates that substantially differ from those of non-overparametrized models. To the best of our knowledge, Xu et al. (2023) is the only work deriving a linear convergence rate for linear networks trained via adaptive step-size GD, but it imposes restrictive step-size constraints, resulting in slow convergence when the initial loss is large.

**Notation.** We use lower case letters $a$ to denote a scalar, and capital letters $A$ and $A^\top$ to denote a matrix and its transpose. We use $\sigma_{\max}(A)$ and $\sigma_{\min}(A)$ to denote the largest and smallest singular values of $A$, $\|A\|_F$ and $\|A\|_2$ to denote its Frobenius and spectral norms, and $A[i, j]$ to denote its $(i, j)$-th element. For a function $f(Z)$, we use $\nabla f(Z) := \frac{\partial}{\partial Z} f(Z)$ to denote its gradient.

## 2 Preliminaries

In this paper, we consider using the GD algorithm to solve the following optimization problem and its overparametrized version

$$\min_{W \in \mathbb{R}^{n \times m}} \ell(W), \qquad \text{(Problem 1)}$$

$$\min_{W_1 \in \mathbb{R}^{n \times h}, W_2 \in \mathbb{R}^{m \times h}} L(W_1, W_2) = \ell(W_1 W_2^\top). \qquad \text{(Problem 2)}$$

We are mostly interested in solving Problem 2, which covers many problems, such as matrix factorization (Koren et al., 2009), matrix sensing (Chen & Chi, 2013), training linear neural networks (Arora et al., 2018; Du et al., 2018a; Xu et al., 2023). In particular, when $\ell(W) = \frac{1}{2}\|Y - XW\|_F^2$, where $X, Y$ are data matrices, Problem 2 corresponds to training a two-layer linear neural network with $n$ inputs, $h$ hidden neurons, $m$ outputs, and weight matrices $W_1$ and $W_2$ using the squared loss.

## 2.1 Convergence rate of GD for Problem 1

In this section, we review the analysis for deriving the convergence rate of GD for Problem 1.

We seek to derive the convergence rate of GD for Problem 1 with the following iterations,

$$W(t+1) = W(t) - \eta_t \nabla \ell(W(t)), \tag{2}$$

where we will use $\ell(t), \nabla \ell(t)$ as a shorthand for $\ell(W(t)), \nabla \ell(W(t))$ respectively.

Throughout the paper, we make the following assumptions.

**Assumption 2.1.** *The loss $\ell(W)$ is twice differentiable, $K$-smooth, and $\mu$-strongly convex .*

**Assumption 2.2.** $\min_W \ell(W) = 0$ .

Assumption 2.1 ensures the solution to Problem 1 is unique. Moreover, commonly used loss functions such as the squared loss and the logistic loss with $\ell_2$ regularization both satisfy Assumption 2.1. Assumption 2.2 is for the purpose of convenience and brevity of theorems in this work. This assumption can be relaxed (to have arbitrary $\ell^*$) without affecting the significance of our results. Moreover, one can have the following inequalities based on the above assumptions for arbitrary $W, V \in \mathbb{R}^{n \times m}$

$$\ell(V) \leq \ell(W) + \langle \nabla \ell(W), V - W \rangle + \frac{K}{2} \|V - W\|_F^2 \qquad \text{Smoothness inequality ,} \tag{3}$$

$$\frac{1}{2} \|\nabla \ell(W)\|_F^2 \geq \mu \ell(W) \qquad \text{PL inequality .} \tag{4}$$

Since strong convexity implies PL condition, equation 4 holds under Assumption 2.1. In §3, we derive the convergence rate of Problem 2 based on the argument of the local PL condition. To be consistent, we highlight the role of the PL condition here. Moreover, the analysis in §2.1 remains applicable when $\mu$-strong convexity is relaxed to $\mu$-PL condition.

In (Polyak, 1963; Boyd & Vandenberghe, 2004), it was shown that whenever $0 < \eta_t < \frac{2}{K}$, the GD iteration equation 2 achieves linear convergence. The derivation is based on two ingredients: Descent lemma and the PL inequality where Descent lemma is derived from the smoothness inequality.

**Descent lemma.** Starting from the smoothness inequality in equation 3, one can substitute $(V, W)$ with the GD iterates $(W(t+1), W(t))$ to derive Descent lemma, i.e.,

$$\ell(t+1) \leq \ell(t) + \langle \nabla \ell(t), W(t+1) - W(t) \rangle + \frac{K}{2} \|W(t+1) - W(t)\|_F^2 = \ell(t) - (\eta_t - \frac{K\eta_t^2}{2}) \|\nabla \ell(t)\|^2 .$$

Based on the PL inequality in equation 4 and Descent lemma above, one can see there is a strict decrease in the loss at each GD step

$$\ell(t+1) \leq \ell(t) - (\eta_t - \frac{K\eta_t^2}{2}) \|\nabla \ell(t)\|^2 \leq (1 - 2\mu\eta_t + \mu K \eta_t^2)\ell(t) , \tag{5}$$

where the fact that $0 < \eta_t < \frac{2}{K}$, implies $0 < 1 - 2\mu\eta_t + \mu K \eta_t^2 < 1$. Moreover, the minimum descent rate in equation 5 is achieved when $\eta_t = \frac{1}{K}$, leading to the following linear convergence rate:

$$\ell(t+1) \leq \left(1 - \frac{\mu}{K}\right) \ell(t) \leq \left(1 - \frac{\mu}{K}\right)^{t+1} \ell(0). \tag{6}$$

**Tightness of the analysis.** The previous analysis guarantees a linear convergence rate for any arbitrary non-convex function that is $K$-smooth and satisfies the $\mu$-PL condition. Moreover, one can show that the rate in equation 6 is optimal in the sense that there exists a function that is $K$-smooth and satisfies the $\mu$-PL condition for which the bound on equation 6 is met with equality. Therefore, one would be tempted to apply such an analysis to Problem 2. We will next show that overparameterization introduces several challenges that prevent this analysis from being readily applied.

## 2.2 Challenges in the Analysis of Convergence of Problem 2 optimized via GD

In this section, we first introduce GD with adaptive step size to solve Problem 2. Then, we discuss the main challenges in deriving the convergence rate for Problem 2 based on the analysis in §2.1.

**Overparametrized GD.** We consider using GD with adaptive step size $\eta_t$ to solve Problem 2

$$\begin{bmatrix} W_1(t+1) \\ W_2(t+1) \end{bmatrix} = \begin{bmatrix} W_1(t) \\ W_2(t) \end{bmatrix} - \eta_t \nabla L(W_1(t), W_2(t)), \tag{7}$$

where $\nabla L(W_1, W_2)$ is computed via the chain rule:

$$\nabla L(W_1, W_2) = \mathcal{T}(\nabla \ell(W); W_1, W_2) := \begin{bmatrix} \nabla \ell(W) W_2 \\ \nabla \ell(W)^\top W_1 \end{bmatrix}. \tag{8}$$

Here $\mathcal{T} : \mathbb{R}^{n \times m} \mapsto \mathbb{R}^{(n+m) \times h}$ is a weight-dependent linear operator that acts on $\nabla \ell(W)$. Thus, the gradient of $L$ in equation 8 can be viewed as a "skewed/scaled gradient" of $\ell$ that depends on $W_1, W_2$. It is this dependence on the weights $W_1, W_2$ that makes it impossible to globally guarantee that equation 3 and equation 4 hold, as shown next.

**Proposition 2.1** (Non-existence of global PL constant and smoothness constant). *Under mild assumptions, the PL inequality and smoothness inequality can only hold globally with constants $\mu_{over} = 0$ and $K_{over} = \infty$ for $L(W_1, W_2)$.*

The proof of the above proposition can be found in Appendix B. Moreover, we present a simple example in Appendix B to help the readers visually understand Proposition 2.1.

The non-existence of global PL and smoothness constants in the over-parametrized models prevents us from using the same proof technique in §2.1 to derive the linear convergence of GD. In §3, we show that although these constants do not exist globally, we can characterize them along iterates of GD. Moreover, under proper choices of the step size of GD, the PL and smoothness constants can be controlled for all iterates of GD. Thus, the linear convergence of GD can be derived.

## 3 Convergence of GD for Problem 2

To deal with the challenges presented in §2.2, in §3.1 we propose a novel PL inequality and Descent Lemma evaluated on the iterates of GD for Problem 2, and show that the local rate of decrease per iteration for Problem 2 is worsened by the condition number of $\mathcal{T}$ compared with the convergence rate of Problem 1. Next, based on the results in §3.1, in §3.2 we show that the condition number of $\mathcal{T}$ during the training can be controlled by its initial value, which helps us deriving a convergence rate for GD that depends on the condition number of $\mathcal{T}$ at initialization, the step size, $K$, and $\mu$. Moreover, we present a sketch of the proof of Theorem 3.2 to highlight the technical novelty and implications of the theorem in §3.3. Finally, in §3.4 and §3.5, we discuss how initialization and width influence the convergence rates derived in Theorem 3.2.

Throughout the paper, we assume that the width satisfies $h \geq \min\{n, m\}$. This assumption ensures $\ell^* = L^*$ where $L^* = \min_{W_1, W_2} L(W_1, W_2)$, and thus solving Problem 2 yields the solution to Problem 1. When $h < \min\{n, m\}$, Problem 2 enforces a rank constraint on the product. Thus, $\min_W \ell(W)$ may not be equal to $\min_{W_1, W_2} L(W_1, W_2)$. We are therefore interested in studying Problem 2 under the assumption $h \geq \min\{n, m\}$ which is the same setting in (Arora et al., 2018; Du et al., 2018a; Xu et al., 2023).

### 3.1 Local PL Inequality and Descent Lemma for Over-parametrized GD

In §2.2, we saw that there does not exist a global PL constant or a global smoothness constant for Problem 2. However, to prove that GD converges linearly to a global minimum of Problem 2, it is sufficient for Descent lemma and PL inequality to hold for iterates of GD. The following theorem formally characterizes the local PL inequality and Descent lemma for Problem 2.

**Theorem 3.1** (Local Descent Lemma and PL condition for GD). *At the $t$-th iteration of GD applied to the Problem 2, Descent lemma and PL inequality hold with local smoothness constant $K_t$ and PL constant $\mu_t$, i.e.,*

$$L(t+1) \leq L(t) - \left(\eta_t - \frac{K_t \eta_t^2}{2}\right)\|\nabla L(t)\|_F^2, \quad \frac{1}{2}\|\nabla L(t)\|_F^2 \geq \mu_t L(t). \tag{9}$$

*Moreover, if the step size $\eta_t$ satisfies $\eta_t > 0$ and $\eta_t K_t < 2$, then the following inequality holds*

$$L(t+1) \leq L(t)(1 - 2\mu_t \eta_t + \mu_t K_t \eta_t^2) := L(t)\rho(\eta_t, t), \tag{10}$$

*where*

$$\mu_t = \mu \sigma_{\min}^2(\mathcal{T}_t), \tag{11}$$

$$K_t = K\sigma_{\max}^2(\mathcal{T}_t) + \sqrt{2KL(t)} + 6K^2\sigma_{\max}(W(t))L(t)\eta_t^2 + 3K\sigma_{\max}^2(\mathcal{T}_t)\sqrt{2KL(t)}\eta_t, \tag{12}$$

*and we use $L(t)$ and $\mathcal{T}_t$ as shorthands for $L(W_1(t), W_2(t))$ and $\mathcal{T}(\,\cdot\,; W_1(t), W_2(t))$, resp.*

The proof of the above theorem can be found in Appendix D. Notice that $\mu_t, K_t$ are not actually constants since they vary w.r.t. the iteration index $t$. In this work, we adopt the convention to call them local PL and smoothness constants to be consistent with the terminology in §2.

In §2.1, we showed that as long as one chooses $\eta_t = \bar{\eta}$, with $0 < \bar{\eta} < \frac{2}{K}$, GD in equation 2 for Problem 1 achieves linear convergence, with an optimal rate $(1 - \frac{\mu}{K})$ given when $\eta_t = \frac{1}{K}$. However, we argue Theorem 3.1 does not imply linear convergence of overparametrized GD even though there always exists sufficiently small $\eta_t > 0$ such that $\eta_t K_t < 2$. The difference is due to the fact that $\mu_t$ and $K_t$ are changing w.r.t. the iterations. Specifically, if $\lim_{t \to \infty} \frac{\mu_t}{K_t} = 0$, one has $\lim_{t \to \infty} \inf_{0 < \eta_t < \frac{2}{K_t}} \rho(\eta_t, t) = 1$. Thus, equation 10 does not necessarily imply that the product of the per-iterate descent $\Pi_{l=0}^t \rho(\eta_l, l)$ goes to zero.

**Towards linear convergence.** Nevertheless, if there exists $\eta_t > 0$ that can simultaneously satisfy the constraint $\eta_t K_t < 2$ and the uniformly bound $1 - 2\mu_t \eta_t + \mu_t K_t \eta_t^2 \leq \bar{\rho} < 1$, for all $t$, one can expect the linear convergence

$$L(t+1) \leq \rho(\eta_t, t)L(t) \leq \bar{\rho}L(t) \leq \bar{\rho}^{t+1}L(0). \tag{13}$$

Guaranteeing a uniform bound as in equation 13, requires one to keep track and control the evolution of $W(t)$, $\mathcal{T}_t$, $\eta_t$ and $L(t)$. In the next section, we will address these issues. For the time being, we focus next on how the $\mu_t, K_t$ in Theorem 3.1 depend on the $\mu, K, L(t), \eta_t$ and the current weights.

**Characterization of $\mu_t, K_t$.** Theorem 3.1 shows how overparametrization affects the local PL constant and smoothness constant, i.e., $\mu_t, K_t$, via a time-varying linear operator $\mathcal{T}_t$. Specifically, the PL constant in equation 11 is the PL constant of $\ell(W)$, i.e., $\mu$, scaled by $\sigma_{\min}^2(\mathcal{T}_t)$. Moreover, the smoothness constant in equation 12 consists of two parts. The first one is $K\sigma_{\max}^2(\mathcal{T}_t)$, which represents the smoothness constant of $\ell(W)$, i.e., $K$, scaled by $\sigma_{\max}^2(\mathcal{T}_t)$. The rest of the terms decrease to zero as the loss $L(t)$ approaches zero.

**Effect of overparametrization on optimization.** Equation 10 in Theorem 3.1 characterizes the rate of decrease per iteration of Problem 2 trained via GD. Around global minimum, $\rho(\eta_t, t)$ takes a simplified form, i.e., $\rho(\eta_t, t) = 1 - 2\mu\sigma_{\min}(\mathcal{T}_t)\eta_t + \mu K \sigma_{\min}(\mathcal{T}_t)\sigma_{\max}(\mathcal{T}_t)\eta_t^2$. By minimizing the $\rho(\eta_t, t)$ over $\eta_t$, one can obtain the optimal local rate decrease per iteration

$$\min_{0 < \eta_t < \frac{2}{K_t}} \rho(\eta_t, t) = 1 - \frac{\mu}{K} \cdot \frac{\sigma_{\min}(\mathcal{T}_t)}{\sigma_{\max}(\mathcal{T}_t)} := 1 - \frac{\mu}{K} \cdot \frac{1}{\kappa(\mathcal{T}_t)}, \tag{14}$$

where we use $\kappa(\mathcal{T}_t)$ to denote the condition number of $\mathcal{T}_t$. Compared with the optimal convergence rate of non-overparametrized model in §2.1, i.e., $1 - \frac{\mu}{K}$, the local rate of decrease of overparametrized model is worsened by $\kappa(\mathcal{T}_t)$. Moreover, the local rate of decrease becomes faster as $\mathcal{T}_t$ is well-conditioned.

In the next section, we will show that proper choice of initialization and step sizes $\eta_t$ does indeed lead to linear convergence of overparametrized GD, and a sufficient condition for $\mathcal{T}_t$ to be well-conditioned.

### 3.2 Linear Convergence of Problem 2 with GD

In this section, we first state a theorem which shows that GD in equation 7 converges linearly to a global minimum of Problem 2 (See Theorem 3.2) under certain constraints on $\eta_t$ and the initialization. Then, based on the convergence rate in Theorem 3.2, we propose an adaptive step size scheduler that accelerates the convergence. We refer the reader to Table 2 for the definition of various quantities appearing in this section.

Table 2: Notation

| Symbol | Definition | Description |
|---|---|---|
| $D(t)$ | $W_1^\top(t)W_1(t) - W_2^\top(t)W_2(t)$ | Imbalance: (almost) training-invariant quantity |
| $\lambda_-$ | $\max\big(\lambda_{\max}(-D(0)),\,0\big)$ | Quantities are related to eigenvalues of *imbalance*. They are used to define $\alpha_1, \alpha_2, \beta_1, \beta_2$ |
| $\lambda_+$ | $\max\big(\lambda_{\max}(D(0)),\,0\big)$ | |
| $\underline{\Delta}$ | $\max\big(\lambda_n(D(0)),0\big) + \max\big(\lambda_m(-D(0)),0\big)$ | |
| $\Delta_+$ | $\lambda_+ - \max\big(\lambda_n(D(0)),0\big)$ | |
| $\Delta_-$ | $\lambda_- - \max\big(\lambda_m(-D(0)),0\big)$ | |
| $\alpha_1$ | $\dfrac{-\Delta_+ - \Delta_- + \sqrt{(\Delta_+ + \underline{\Delta})^2 + 4\beta_1^2} + \sqrt{(\Delta_- + \underline{\Delta})^2 + 4\beta_1^2}}{2}$ | Lower bound on $\sigma_{\min}^2(\mathcal{T}_t)$ |
| $\alpha_2$ | $\dfrac{\lambda_+ + \sqrt{\lambda_+^2 + 4\beta_2^2}}{2} + \dfrac{\lambda_- + \sqrt{\lambda_-^2 + 4\beta_2^2}}{2}$ | Upper bound on $\sigma_{\max}^2(\mathcal{T}_t)$ |
| $\beta_1$ | $\max\left(0,\ \sigma_{\min}(W^*) - \sqrt{\tfrac{K}{\mu}}\,\|W(0) - W^*\|_F\right)$ | Lower bound on $\sigma_{\min}\big(W_2(t)W_1(t)\big)$ |
| $\beta_2$ | $\sigma_{\max}(W^*) + \sqrt{\tfrac{K}{\mu}}\,\|W_1(0)W_2^\top(0) - W^*\|_F$ | Upper bound on $\sigma_{\max}\big(W_2(t)W_1(t)\big)$ |

We now present our main result on the linear convergence of GD for Problem 2.

**Theorem 3.2** (Linear convergence of GD for Problem 2). *Suppose $\ell$ satisfies Assumption 2.1 and Assumption 2.2, and given $h \geq \min(m,n)$, we assume GD in equation 7 is initialized so that $\alpha_1 > 0$. Then there exists $\eta_{\max} > 0$ such that $\forall \eta_0, \eta_t$ that satisfies $0 < \eta_0 < \eta_{\max}$ and*

$$\eta_0 \leq \eta_t \leq \min\big((1+\eta_0^2)^{\frac{t}{2}}\eta_0,\ \frac{1}{K_t}\big), \tag{15}$$

*one can derive the following bound for each iteration: $\bar{\mu} \leq \mu_t \leq K_t \leq \bar{K}_t$, and*

$$L(t+1) \leq L(t)\rho(\eta_t, t) \leq L(t)\bar{\rho}(\eta_0, 0). \tag{16}$$

*Moreover, based on equation 16, GD algorithm in equation 7 converges linearly*

$$L(t+1) \leq L(0)\bar{\rho}(\eta_0, 0)^{t+1}, \tag{17}$$

*where*

$$\bar{\rho}(\eta_t, t) = 1 - 2\bar{\mu}\eta_t + \bar{\mu}\bar{K}_t\eta_t^2, \quad \bar{\mu} = \mu\big[\alpha_1 + 2\alpha_2\big(1 - \exp(\sqrt{\eta_0})\big)\big], \quad \Delta = (1+\eta_0^2)\bar{\rho}(\eta_0, 0),$$
$$\bar{K}_t = \sqrt{2KL(0)\bar{\rho}(\eta_0,0)^t} + 6K^2\beta_2 L(0)\eta_0^2\Delta^t + K\exp(\sqrt{\eta_0})\alpha_2\big[1 + 3\sqrt{2KL(0)\Delta^t}\eta_0\big].$$

The proof of the above theorem is presented in Appendix E. The above theorem states GD enjoys linear convergence for Problem 2 under the assumptions that $\alpha_1 > 0$ and certain constraints on $\eta_t$. We make the following remarks:

**Conditions on the initialization for linear convergence.** From Theorem 3.2, we see that if the initialization $\{W_1(0), W_2(0)\}$ satisfies $\alpha_1 > 0$, then GD converges linearly with an appropriate choice of the step size. The constraints on $\eta_0$ ensure that $\bar{\mu} > 0$ and $0 < \bar{\rho}(\eta_0, 0) < 1$. The assumptions on $\alpha_1$ has been studied in Min et al. (2022) where the authors show that $\alpha_1 > 0$ when there is either 1) sufficient

imbalance $\underline{\Delta} > 0$ or 2) sufficient margin $\beta_1 > 0$, where $\underline{\Delta}, \beta_1$ is defined in Table 2. In §3.4, we present two conditions that ensure $\alpha_1 > 0$ which covers commonly used Gaussian initialization, Xavier initialization, and He initialization. Please see §3.4 for a detailed proof and discussions.

**Evolution of smoothness constant.** One unique feature in our Theorem is the time-varying upper bound $\bar{K}_t$ on the local smoothness constant $K_t$ along GD iterates. The constraints on $\eta_0$ ensures that $0 < \bar{\rho}(\eta_0, 0), \Delta < 1$. Thus, $\bar{K}_t$ monotonically decrease to $K \exp(\sqrt{\eta_0})\alpha_2$ w.r.t. $t$. The fact that $\bar{K}_t$ is monotonically decreasing w.r.t. $t$ suggests that the local optimization landscape gets more benign as the training proceeds. Thus in order to achieve a fast rate of convergence, there is a need for a time-varying choice of step size that adapts to the changes in the local smoothness constant $K_t$ (because theoretically, the optimal choice is $\eta_t = \frac{1}{K_t}$, based on equation 10).

**Requirement on the step size.** We have mentioned in the previous remark that a time-varying step size could be beneficial for convergence. However, prior analyses (Arora et al., 2018; Du & Hu, 2019; Xu et al., 2023) are all restricted to a constant or decaying step size. The main reason is that one requires a uniform spectral bound on $\mathcal{T}_t$ and $W(t)$ throughout the entire GD trajectory to establish linear convergence and such a uniform bound has only been shown under a constant or decaying step size. In our analysis, we show a similar spectral bound can be obtained even with a growing step size (See Appendix E), as long as $\eta_t \le (1 + \eta_0^2)^{\frac{t}{2}}\eta_0$, but not too much $\eta_t \le \frac{1}{K_t}$ (ensures a sufficient decrease in the loss at every iteration). The first bound diverges to infinity exponentially fast, and the second bound has a growing lower bound $\frac{1}{\bar{K}_t}$ which monotonically increases to $\frac{1}{K\exp(\sqrt{\eta_0})\alpha_2}$. Thus, initially, the step size is restricted to $[\eta_0, (1 + \eta_0^2)^{\frac{t}{2}}\eta_0]$. As the training goes on, the binding constraint becomes $\eta_t \le \frac{1}{\bar{K}_t}$, suggesting that GD can take a step that achieves the theoretically largest descent in the loss. In Davis et al. (2024), the authors study GD with carefully designed adaptive step sizes (GDPolyak) applied to loss functions that exhibit quartic growth away from the solution set. Their analysis covers rank-overparameterized matrix sensing problems and student-teacher setups involving a single neuron. GDPolyak differs from our method in how the step sizes are determined. In our approach, the step size at each iteration follows a classic optimization strategy: we estimate the next iterate's loss based on the current loss and weights (see equation 16), then choose the step size by minimizing this upper bound (see equation 18). As a result, our adaptive schedule requires accurate characterization of the local PL constant and local smoothness constant. By contrast, GDPolyak alternates between constant step sizes and Polyak step sizes, relying on the latter to adapt to local features of the loss landscape.

**Local rate of convergence .** In Theorem 3.2, we show $L(t+1) \le L(t)\rho(\eta_t, t)$ and $\bar{\mu} \le \mu_t \le K_t \le \bar{K}_t$ when $\eta_t$ satisfies certain constraints. When $t$ is sufficiently large, or equivalently around any global minimum of Problem 2, the optimal rate of convergence is achieved as (via a proper choice of $\eta_t$)

$$\min_{\eta_t} \rho(\eta_t, t) = 1 - \frac{\mu}{K} \cdot \frac{\mu_t}{K_t} \le 1 - \frac{\mu}{K} \cdot \frac{\alpha_1 + 2\alpha_2(1 - \exp(\sqrt{\eta_0}))}{\exp(\sqrt{\eta_0})\alpha_2} .$$

Notice the optimal local rate of convergence can be arbitrarily close to $1 - \frac{\mu}{K} \cdot \frac{\alpha_1}{\alpha_2}$ as $\eta_0$ decreases. Moreover, compared with the optimal convergence rate of the non-overparametrized model, the optimal local rate of convergence of the overparametrized model is worsened by $\frac{\alpha_1}{\alpha_2}$, which is an upper bound on $\kappa(\mathcal{T}_t)$. In §3.5, we provide a sufficient condition for $\mathcal{T}_0$ to be well-conditioned, i.e. $\frac{\alpha_1}{\alpha_2}$ to be close to one, and numerically verfity this condition in AppendixG.

**Comparison in local rate of convergence with SOTA.** We compare our results with prior works studying the same problem Arora et al. (2018); Du et al. (2018a); Xu et al. (2023) (See Table 3). Moreover, we present a detailed discussion on the difference between proof techniques used in this work and prior work, and how it leads to different convergence rates. Please see Appendix C for details.

**Choices of the step size.** Recall that for non-overparamterized GD, we have $\ell(t+1) \le (1 - 2\mu\eta_t + \mu K \eta_t^2)\ell(t)$, there exists an optimal choice of $\eta_t^* = \frac{1}{K}$ that minimize the theoretical upper bound on $\ell(t+1)$. In Theorem 3.1 and Theorem 3.2, we show $L(t+1) \le h(\eta_t, t)L(t)$ under certain conditions on $\eta_t$ where $h(\eta_t, t) \in \{\rho(\eta_t, t), \bar{\rho}(\eta_t, t)\}$. It is natural to use a similar approach to select step size at each iteration. To achieve the optimal step size, it suffices to minimize the upper bound on $L(t+1)$ to achieve the most decrease

Table 3: Comparison of convergence rates between prior work and our work.

|  | loss | step size | initialization | local rate of convergence |
|---|---|---|---|---|
| Arora et al. (2018) | squared loss | constant | $D(0) \approx 0, \beta_1 > 0$ | $1 - \Omega(\frac{\mu \alpha_1^2}{K \alpha_2^2})$ |
| Du et al. (2018a) | squared loss | decreasing | $D(0) \approx 0, \beta_1 > 0$ | no explicit rate |
| Xu et al. (2023) | squared loss | constant | $\alpha_1 > 0$ | $1 - \Omega(\frac{\mu \alpha_1^2}{K \alpha_2^2})$ |
| our work | general | adaptive | $\alpha_1 > 0$ | $1 - \Omega(\frac{\mu \alpha_1}{K \alpha_2})$ |

at each iteration. The difference is that we have a time-varying upper bound on $L(t+1)$ thus the minimizer $\eta_t^*$ depends on time, and our choice of $\eta_t^*$ must respect our constraint on step size in equation 15. This leads to the following choice for $\eta_t^*$:

$$\eta_t^* = \underset{\eta_t \le \min\{(1+\eta_0^2)^{t/2}\eta_0, \frac{1}{K_t}\}}{\arg\min} h(\eta_t, t) \,. \tag{18}$$

Since $\rho(\eta_t, t), \bar{\rho}(\eta_t, t)$ are quadratic in terms of $\eta_t$, so $\eta_t^*$ takes the following closed-form solutions depending on which upper bound to use:

$$\eta_t^* = \begin{cases} \min\big((1+\eta_0^2)^{t/2}\eta_0, \frac{1}{K_t}\big) & \text{if } h(\eta_t) = \rho(\eta_t, t) \,, \\ \min\big((1+\eta_0^2)^{t/2}\eta_0, \frac{1}{K_t}\big) & \text{if } h(\eta_t) = \bar{\rho}(\eta_t, t) \,. \end{cases} \tag{19}$$

The above choices of the adaptive step sizes satisfy the constraints in Theorem 3.2, so they both guarantee linear convergence for over-parametrized GD. Moreover, such choices of $\eta_t$ give us the following theoretical bound on $L(t+1)$, i.e.,

$$L(t+1) \le L(0) \prod_{k=1}^{t} h(\eta_k^*, k) \,. \tag{20}$$

In §4.1, we provide numerical verification of the close alignment between the theoretical bounds stated above and the actual convergence rate. We also observe an accelerated convergence when employing the step sizes specified in equation 18 compared with the one proposed in Xu et al. (2023) and Backtracking line search. We refer the readers to §4 for simulation results.

### 3.3 Proof Sketch

In this section, we present a proof sketch of Theorem 3.2 that highlights the key technical contributions. In Theorem 3.1, we established a local PL inequality and a local Descent Lemma. Moreover, one can show the upper bound

$$L(t+1) \le L(t) \left(1 - 2\mu_t \eta_t + \mu_t K_t \eta_t^2\right) = L(t) \rho(\eta_t, t) \,. \tag{21}$$

As discussed in §3.1, this inequality alone does not guarantee linear convergence. For instance, if $\lim_{t\to\infty} \frac{\mu_t}{K_t} = 0$, then $\lim_{t\to\infty} \rho(\eta_t, t) = 1$. To show there exists a $0 < \bar{\rho} < 1$ such that $\rho(\eta_t, t) \le \bar{\rho}$ holds for all $t$, we use the following two-step approach in a similar way as it was done in Xu et al. (2023).

**Step one: uniform spectral bounds for $\mathcal{T}_t$ and $W_t$ .** First, we show when $\eta_t$ is controlled, one can have the uniform spectral bounds on $\mathcal{T}_t$ and $W(t)$. The following lemma characterizes this property formally.

**Lemma 3.1** (Uniform spectral bounds on $\mathcal{T}_t, W(t)$.)**.** *Under the same assumption and constraints in Theorem 3.2, one has the following uniform spectral bounds on $\mathcal{T}_t, W(t)$*

$$\alpha_1 + 2\alpha_2\big(1 - \exp(\sqrt{\eta_0})\big) \le \sigma_{\min}^2(\mathcal{T}_t) \le \sigma_{\max}^2(\mathcal{T}_t) \le \alpha_2 \exp(\sqrt{\eta_0}) \,, \tag{22}$$

$$\beta_1 \le \sigma_{\min}(W(t)) \le \sigma_{\max}(W(t)) \le \beta_2 \,. \tag{23}$$

The above lemma shows the uniform spectral bounds on $\mathcal{T}_t$ and $W_t$ depending on $\alpha_1$, $\alpha_2$, $\beta_1$, $\beta_2$ and $\eta_0$. Moreover, the bounds on the singular values of $\mathcal{T}_t$ can be arbitrarily close to $\alpha_1, \alpha_2$ as $\eta_0$ approaches zero.

Similar results have been derived in Xu et al. (2023) where the authors show uniform spectral bounds of $\mathcal{T}_t, W_t$ for constant step size GD. Our proof strategy is similar to Xu et al. (2023) which relies on the fact that when the GD enjoys linear convergence, the change of imbalance during the training is small. For constant step size GD, we can characterize the change using the step size $\eta$ and the convergence rate $\bar{\rho}$ i.e. $\|D(t) - D(0)\|_F \leq \mathcal{O}(\frac{\eta^2}{1-\bar{\rho}})$. In this work, we discover when we allow step size to grow but not too fast, i.e. $\eta_t \leq (1+\eta_0^2)^{\frac{t}{2}}\eta_0$, we still can control the change of imbalance, i.e. $\|D(t) - D(0)\|_F \leq \mathcal{O}(\frac{\eta_0^2}{1-\Delta})$ where $\Delta = (1+\eta_0^2)\bar{\rho}(\eta_0, 0)$. This observation helps us derive the uniform spectral bounds for $\mathcal{T}_t, W(t)$ while allowing the step size to grow. The bound on the change of imbalance is not restricted to scenarios when loss converges linearly. In Ghosh et al. (2025), the authors demonstrate that the imbalance gap decreases at a linear rate, even in the edge-of-stability regime. We mention this work here as an additional reference for readers interested in such phenomena.

**Step two.** Second, we employ an induction-based argument to show that based on Lemma 3.1, one can show $\mu \geq \bar{\mu}, K_t \leq \bar{K}_t$ and $L(t)$ converges linearly with the rate $\bar{\rho}_0$.

**Lemma 3.2** (Induction step to show $\mu_t, K_t$ is bounded and $L(t)$ converges linearly.)**.** *Under the same assumption and constraints in Theorem 3.2, assume $L(t)$ enjoys linear convergence with rate $\bar{\rho}(\eta_0, 0)$ until iteration $k$, then the following holds for iteration $k+1$*

$$\mu_{k+1} \geq \bar{\mu}, \quad K_{k+1} \leq \bar{K}_{k+1}, \tag{24}$$

*with $\bar{\mu}, \bar{K}_{k+1}$ defined in Theorem 3.2. Moreover, one can show*

$$\rho(\eta_{k+1}, k+1) \leq \bar{\rho}(\eta_{k+1}, k+1) \leq \bar{\rho}(\eta_0, 0). \tag{25}$$

Equation 24 is a direct consequence of Lemma 3.1 and the induction that $L(t)$ enjoys linear convergence until iteration $k$. We can lower bound $\mu_k$ by subsituting $\sigma_{\min}(\mathcal{T}_k)$ with the bound in equation 22 and upper bound $K_t$ by subsituting $\sigma_{\max}(\mathcal{T}_k), \sigma_{\max}(W(k)), L(k)$ with the bound in equation 22, equation 23 and $L(0)\bar{\rho}(\eta_0, 0)^t$ respectively. Based on these results, one can derive the following upper bound on $L(k+1)$ under the same constraints on $\eta_t$ in Theorem 3.2

$$L(k+1) \leq L(k) - \left(\eta_k - \frac{K_k\eta_k^2}{2}\right)\|\nabla L(k)\|_F^2 \qquad \textbf{Local Descent lemma in Theorem 3.1} \tag{26}$$

$$\leq L(k) - 2\bar{\mu}\left(\eta_k - \frac{K_k\eta_k^2}{2}\right)L(k) \qquad \textbf{Local PL inequality with } \bar{\mu} \tag{27}$$

$$= (1 - 2\bar{\mu}\eta_k + \bar{\mu}K_k\eta_k^2)L(k) \qquad \textbf{Use } K_k \leq \bar{K}_k \leq \bar{K}_0 \tag{28}$$

$$\leq (1 - 2\bar{\mu}\eta_k + \bar{\mu}\bar{K}_0\eta_k^2)L(k) \qquad \textbf{Use constraints on } \eta_0 \textbf{ and equation 15} \tag{29}$$

$$\leq \bar{\rho}(\eta_0, 0)L(k). \tag{30}$$

In summary, we show GD in equation 7 achieves linear convergence with the rate $\bar{\rho}(\eta_0, 0)$ under the constraints on $\eta_t$ and the assumption $\alpha_1 > 0$ in this section. Moreover, we show the local rate of convergence depends on $\frac{\alpha_1}{\alpha_2}$. In the next section, we theoretically show that commonly used random initialization leads to $\alpha_1 > 0$, and large width and large initialization scale leads to faster local rate of convergence, i.e., $\frac{\alpha_1}{\alpha_2} \to 1$.

### 3.4 Proper initialization ensures $\alpha_1 > 0$

In §3.2, we see that under the assumption $\alpha_1 > 0$, Problem 2 trained via GD in equation 7 converges linearly to a global minimum under certain constraints on the step sizes. In this section, we present two conditions on the initialization that ensure $\alpha_1 > 0$.

We first show two conditions on the initialization that ensure $\alpha_1 > 0$.

**Lemma 3.3** (Mild overparametrization ensures $\alpha_1 > 0$)**.** *Let $W_1(0), W_2(0)$ are initialized entry-wise i.i.d. from a continuous distribution $\mathbb{P}$. When $h \geq m + n$, $\alpha_1 > 0$ holds almost surely over random initialization with $W_1(0), W_2(0)$.*

**Lemma 3.4** (Lemma 1 in (Min et al., 2021)). *Let $W_1(0), W_2(0)$ are initialized entry-wise i.i.d. from $\mathcal{N}(0, \frac{1}{h^{2p}})$ with $\frac{1}{4} \leq p \leq \frac{1}{2}$. For $\forall \delta > 0$ and $h \geq poly(n, m, \frac{1}{\delta})$, with probability $1 - \delta$ over random initialization with $W_1(0), W_2(0)$, the following holds $\alpha_1 \geq h^{1-2p}$.*

We refer the readers to Appendix F for detailed proof. Both lemmas presented above ensure $\alpha_1 > 0$ under different conditions on the width. Compared with Lemma 3.4, Lemma 3.3 considers a wider range of distributions that include Gaussian distribution and uniform distribution. Thus, commonly used random initialization schemes, such as Xavier initialization (Glorot & Bengio, 2010) and He initialization (He et al., 2015), lead to $\alpha_1 > 0$. Moreover, the requirement of overparametrization in Lemma 3.3 is mild compared with the one in Lemma 3.4, i.e., $h \geq m + n$ versus $h \geq poly(n, m, \frac{1}{\delta})$. As a result, Lemma 3.3 can be applied to more general overparametrization. On the other hand, the conclusion of Lemma 3.3 is weaker than Lemma 3.4 in the sense that Lemma 3.3 only proves $\alpha_1 > 0$ but does not characterize its magnitude while Lemma 3.4 characterizes the lower bound on $\alpha_1$ will increase as $h$ increases.

### 3.5 Large width and proper choices of the variance lead to well-conditioned $\mathcal{T}_0$

In this section, we show that a proper choice of initialization and width can lead to well-conditioned $\mathcal{T}_0$, i.e., $\frac{\alpha_1}{\alpha_2} \to 1$. Based on the results in §3.2, we show that the local convergence rate of overparametrized model trained via GD can match the rate of the non-overparametrized model.

**Theorem 3.3.** *Let $W_1(0), W_2(0)$ are initialized entry-wise i.i.d. from $\mathcal{N}(0, \frac{1}{h^{2p}})$ with $\frac{1}{4} < p < \frac{1}{2}$. $\forall \delta \in (0, 1), h \geq poly(m, n, \frac{1}{\delta})$, with probability $1 - \delta$ over random initialization $W_1(0), W_2(0)$, the following holds*

$$\frac{\sigma_{\min}(\mathcal{T}_0)}{\sigma_{\max}(\mathcal{T}_0)} \geq \frac{\alpha_1}{\alpha_2} \geq 1 - \Omega(h^{2p-1}). \tag{31}$$

We refer the readers to Appendix F.1 for detailed proof. The above theorem states the condition number of $\mathcal{T}_0$ approaches one when the width increases to infinity under suitable choices of the variance of Gaussian initialization. Therefore, increasing the width can lead to a fast convergence rate.

In Appendix G, we will numerically show that with proper choices of the variance of the initialization and the adaptive step size proposed in §3.2, a large width will lead to well-conditioned $\mathcal{T}_t$ and the convergence rate of the overparametrized model can asymptotically match the rate of the non-overparametrized model.

## 4 Experiments

In this section, we first present empirical evidence that Theorem 3.2 provides a good characterization of the actual convergence rate under different initialization in §4.1. Then, in §4.2, we compare the convergence rate of GD using the adaptive step size proposed in equation 19, in Section 3.3 of Xu et al. (2023), and backtracking line search. Throughout the experiments, we consider Problem 2 with squared loss

$$L(W_1, W_2) = \frac{1}{2} \|Y - X W_1 W_2^\top\|_F^2, \tag{32}$$

where $X, Y \in \mathbb{R}^{10 \times 10}$ are data matrices and $W_1, W_2 \in \mathbb{R}^{10 \times h}$ are the weights. This can be viewed as a two-layer linear network with input and output dimensions 10 and the width of the hidden layer to be $h$. Throughout the simulations, we choose $h \in \{500, 1000, 4000\}$. We choose $c = 0.5, d = 1.01$ in Theorem E.1. The initialization of the weights and generation of data matrices are as follows: $W_1(0), W_2(0) \in \mathbb{R}^{10 \times h}$, and have entry-wise i.i.d. samples drawn from $\mathcal{N}(0, 1)$. We generate $X$ as a random orthogonal matrix, and $Y = X W_1(0) W_2(0) + \sigma^2 \epsilon$ where $\epsilon \in \mathbb{R}^{10 \times 10}$ and are entry-wise i.i.d. samples drawn from $\mathcal{N}(0, 1)$. When $\sigma^2$ is large, the initial loss is large, thus the margin is small. Moreover, we experimentally observe that the initial imbalance grows w.r.t. $h$. The choices of $h$ and $\sigma$ allow us to test our results in different regimes.

### 4.1 Evaluation of the Tightness of the Theoretical Bound on the Convergence Rate

Figure 1 compares the actual convergence rate of $L(t)$ versus the theoretical upper bound in §3.2 for different choices of $\sigma$ and $h$, and dissimilar $\frac{\alpha_1}{\alpha_2}$. In all cases, the theoretical upper bound follows the actual loss well.

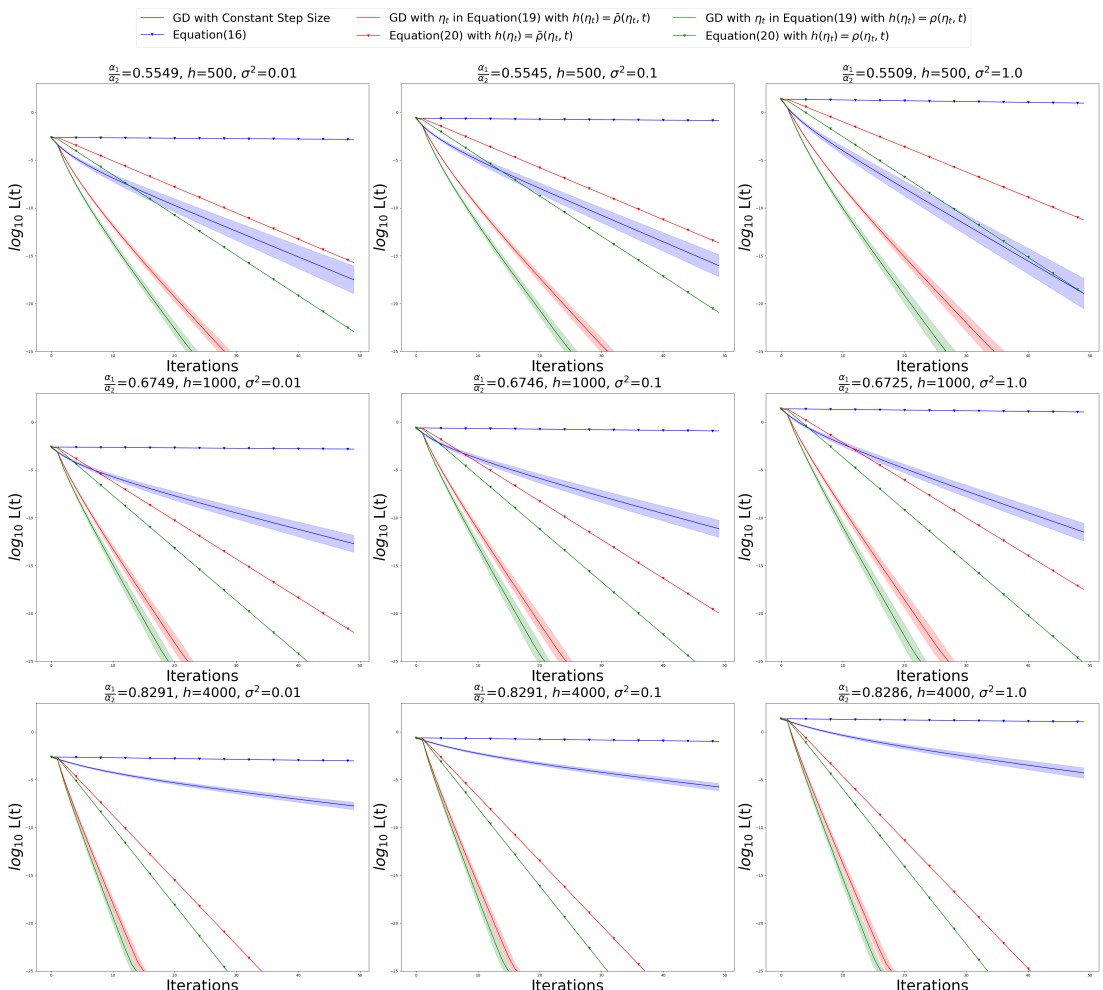

Figure 1: Tightness of the theoretical upper bound versus reconstruction error $L(t)$ for different choices of step size in §3.2, shown in different colors. We run the simulations for nine different settings of initialization and data generation. For each setting, we repeat the simulation thirty times. The triangle lines represent the theoretical upper bound on the training loss in equation 17 and equation 20. The solid lines represent the mean of the $\log_{10}$ of the reconstruction error $L(t)$. The shaded area is the mean of $\log_{10} L(t)$ plus and minus one standard deviation.

Moreover, we observe for each adaptive step size scheme, the theoretical bounds and the actual rate of convergence become slower as $\frac{\alpha_1}{\alpha_2}$ decreases. This is because our bounds on the local rate of convergence depend on $\frac{\alpha_1}{\alpha_2}$, and the smaller $\frac{\alpha_1}{\alpha_2}$, the slower the convergence rate.

## 4.2 Comparison with Prior Work and Backtracking Line Search

In this subsection, we compare the adaptive step sizes proposed in Xu et al. (2023), backtracking line search with the step sizes proposed in equation 19 with $h(\eta_t) = \rho(\eta_t, t)$. We set the hyperparameters of the adaptive step size scheme proposed in Xu et al. (2023) to be $c_1 = 0.5, c_2 = 1.5$, which is the same setting in their simulations. We refer the readers to Appendix G for detailed descriptions of backtracking line search. Figure 2 shows that the step size choice proposed in equation 19 achieves the fastest convergence compared with Xu et al. (2023) and backtracking line search in different settings. This is because for the adaptive step size scheduler in this work, the step size at each iteration has closed form (See equation 19), thus the time for picking the optimal step size per iteration is negligible. The only time-consuming part is to find $\eta_0$ since one needs to solve equation 101 and equation 102 to get $\eta_{\max}$. For the step size proposed in Xu

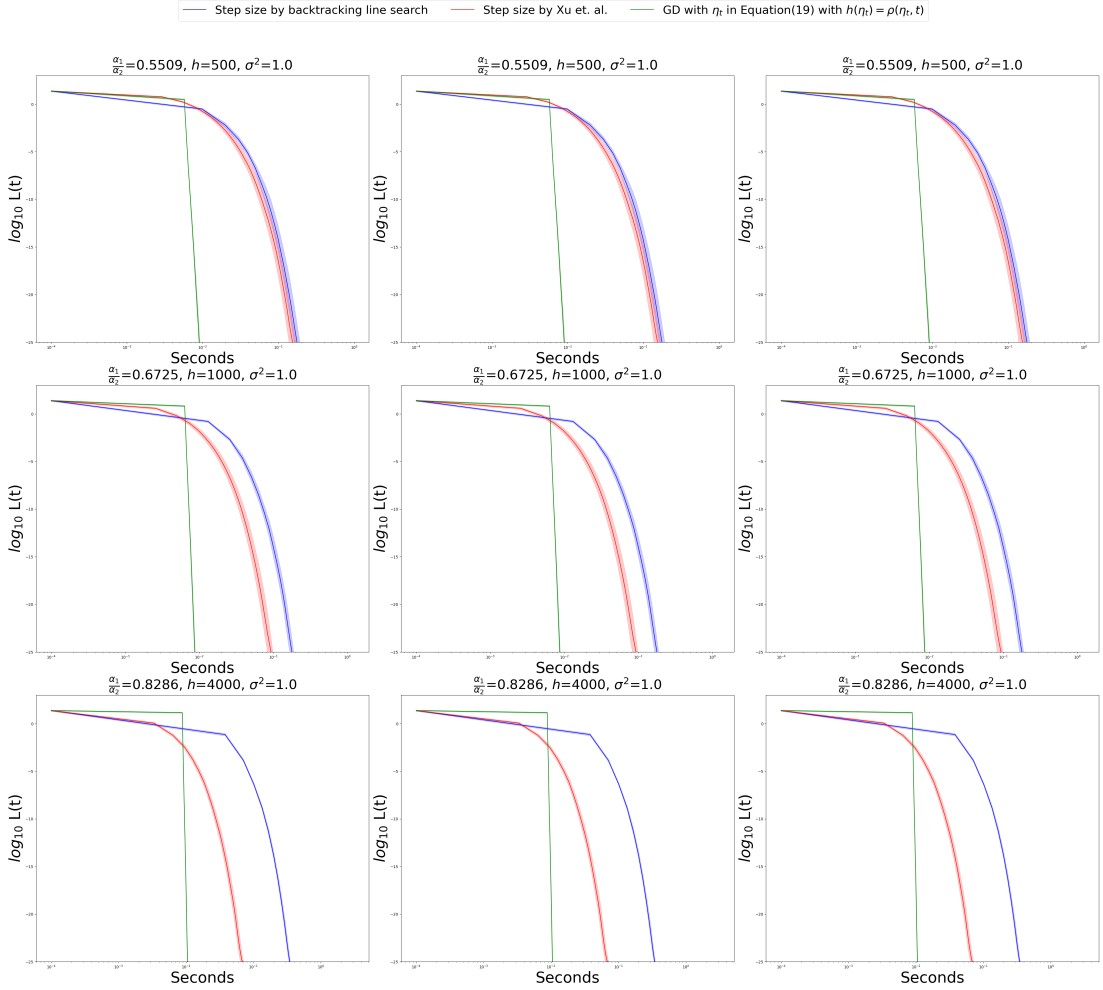

Figure 2: Evolution of the loss and of the step size for different choices of the step size schedule under different initialization and data generation. We run the simulations thirty times. For each setting, we repeat the simulation thirty times. The solid lines represent the mean of $\log_{10}$ of the reconstruction error $L(t)$. The shaded area is the mean of $\log_{10} L(t)$ plus and minus one standard deviation.

et al. (2023), the algorithm consists of solving a third-order polynomial at each iteration, which results in larger computational time. Moreover, the adaptive scheduler proposed in our work follows a sharper characterization of the local convergence rate than Xu et al. (2023), and the adaptive step size scheduler in this work theoretically converges of order $\frac{\alpha_1}{\alpha_2}$ faster than the one proposed in Xu et al. (2023). For the backtracking line search algorithm, since the algorithm iteratively searches for the step size at each iteration. Therefore, the time cost for each iteration is high as well.

## 5   Conclusion

This paper studies the convergence of GD for optimizing two-layer linear networks with general loss functions. Specifically, we derive a linear convergence rate for finite-width networks initialized outside the NTK regime. We use a common framework for studying the convergence of GD for the non-convex optimization problem, i.e. PL condition and Descent lemma. Although the loss landscape of neural networks does not satisfy the PL condition and Descent lemma with global constants, we show that when the step size is small, both conditions satisfy locally with constants depending on the singular value of the weights, the current loss, and the singular value of the products. Furthermore, we prove that the local PL and smoothness

constants can be uniformly bounded by the initial imbalance, margin, PL constant, and smoothness constant of the non-overparameterized model. Finally, we provide an explicit convergence rate dependent on margin, imbalance, and the condition number of the non-overparameterized model. Based on this rate, we propose an adaptive step size scheme that accelerates convergence compared to a constant step size.

### Acknowledgements

The authors acknowledge the support of the Office of Naval Research (grant 503405-78051), the National Science Foundation (grants 2031985, 2212457 and 2330450), and the Simons Foundation (grant 814201).

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

# A   Preliminary lemmas

In this section, we present a preliminary lemma which will be used in the following sections.

**Lemma A.1** (Inequality on the Frobenius norm). *For matrix $A, B, C, D$, we have*

$$\langle A, B \rangle \leq \|A\|_F \cdot \|B\|_F \,, \tag{33}$$

$$2\|AB\|_F \leq \|A\|_F^2 + \|B\|_F^2 \,, \tag{34}$$

$$\|AB + CD\|_F^2 \leq [\sigma_{\max}^2(A) + \sigma_{\max}^2(C)]^2 \cdot [\|B\|_F^2 + \|D\|_F^2] \,, \tag{35}$$

$$\|A\|_F^2 + \|B\|_F^2 \leq 2\|A + B\|_F^2 \,, \tag{36}$$

$$\sigma_{\min}^2(A)\|B\|_F^2 \leq \|AB\|_F^2 \leq \sigma_{\max}^2(A)\|B\|_F^2 \,, \tag{37}$$

$$\sigma_{\min}^2(B)\|A\|_F^2 \leq \|AB\|_F^2 \leq \sigma_{\max}^2(B)\|A\|_F^2 \,. \tag{38}$$

Lemma A.1 has been derived and used multiple times in prior work. We refer the readers to Appendix C in Xu et al. (2023) for detailed proof.

**Lemma A.2** (Singular values of $\mathcal{T}$). *The largest and smallest singular values of $\mathcal{T}$ are given as*

$$\sigma_{\min}^2(\mathcal{T}) = \sigma_{\min}^2(W_1) + \sigma_{\min}^2(W_2) \,,$$
$$\sigma_{\max}^2(\mathcal{T}) = \sigma_{\max}^2(W_1) + \sigma_{\max}^2(W_2) \,. \tag{39}$$

*Proof.* First, one can see

$$\mathcal{T}^* \circ \mathcal{T}(U; W_1, W_2) = UW_2W_2^\top + W_1W_1^\top U \,, \tag{40}$$

where $\mathcal{T}^*$ is the adjoint of $\mathcal{T}$. Then, we use Min-max theorem to show

$$\lambda_{\min}(\mathcal{T}^* \circ \mathcal{T}) = \sigma_{\min}^2(W_1) + \sigma_{\min}^2(W_2) \,, \quad \lambda_{\max}(\mathcal{T}^* \circ \mathcal{T}) = \sigma_{\max}^2(W_1) + \sigma_{\max}^2(W_2) \,. \tag{41}$$

Let the singular value decompositions of $W_1, W_2$ be

$$W_1 = U_1\Sigma_1 V_1^\top = \sum_{i=1}^{r_1} \sigma_{1,i} u_{1,i} v_{1,i}^\top \,, \quad W_2 = U_2\Sigma_2 V_2^\top = \sum_{i=1}^{r_2} \sigma_{2,i} u_{2,i} v_{2,i}^\top \,, \tag{42}$$

where $r_1 = \text{rank}(W_1), r_2 = \text{rank}(W_2)$, and $\{\sigma_{1,i}\}_{i=1}^{r_1}, \{\sigma_{2,i}\}_{i=1}^{r_2}$ are of descending order. Then, one has the following

$$
\begin{aligned}
\lambda_{\min}(\mathcal{T}^* \circ \mathcal{T}) &= \min_{\|U\|_F=1} \langle U, UW_2W_2^\top + W_1W_1^\top U \rangle \\
&= \min_{\|U\|_F=1} \langle U, UW_2W_2^\top \rangle + \min_{\|U\|_F=1} \langle U, W_1W_1^\top U \rangle \\
&\geq \sigma_{\min}^2(W_1) + \sigma_{\min}^2(W_2) \,.
\end{aligned} \tag{43}
$$

On the other hand, if one choose $U = v_{1,r_1} u_{2,r_2}^\top$, the following equation holds

$$
\begin{aligned}
&\langle U, UW_2W_2^\top + W_1W_1^\top U \rangle \\
=&\langle v_{1,r_1} u_{2,r_2}^\top, v_{1,r_1} u_{2,r_2}^\top W_2W_2^\top + W_1W_1^\top v_{1,r_1} u_{2,r_2}^\top \rangle \\
=&\langle v_{1,r_1} u_{2,r_2}^\top, v_{1,r_1} u_{2,r_2}^\top \sum_{i=1}^{r_2} \sigma_{2,i}^2 u_{2,i} v_{2,i}^\top \rangle + \langle v_{1,r_1} u_{2,r_2}^\top, \sum_{i=1}^{r_1} \sigma_{1,i}^2 u_{1,i} v_{1,i}^\top v_{1,r_1} u_{2,r_2}^\top \rangle \\
=&\sum_{i=1}^{r_2} \sigma_{2,i}^2 \text{tr}(u_{2,r_2} v_{1,r_1}^\top v_{1,r_1} u_{2,r_2}^\top u_{2,i} v_{2,i}^\top) + \sum_{i=1}^{r_2} \sigma_{1,i}^2 \text{tr}(u_{2,r_2} v_{1,r_1}^\top u_{1,i} v_{1,i}^\top v_{1,r_1} u_{2,r_2}^\top) \\
=&\sigma_{1,r_1}^2 + \sigma_{2,r_2}^2 \,,
\end{aligned} \tag{44}
$$

where the last line is based on the fact that $v_{1,i}^\top v_{i,r_1} = 0, u_{2,j}^\top u_{2,r_2} = 0$ holds for all $i \neq r_1, j \neq r_2$. Therefore, based on equation 43 and equation 44, one has

$$\lambda_{\min}(\mathcal{T}^* \circ \mathcal{T}) = \sigma_{\min}^2(W_1) + \sigma_{\min}^2(W_2). \tag{45}$$

Similarly, we can show

$$\lambda_{\max}(\mathcal{T}^* \circ \mathcal{T}) = \sigma_{\max}^2(W_1) + \sigma_{\max}^2(W_2). \tag{46}$$

$\square$

**Lemma A.3** (Singular values of random matrix). *Given $m, n \in \mathbb{N}$ with $m \leq n$. Let $A \in \mathbb{R}^{n \times m}$ be a random matrix with i.i.d. standard normal entries. For any $\delta > 0$, with probability at least $1 - 2\exp(\delta^2)$, one has*

$$\sqrt{n} - \sqrt{m} - \delta \leq \sigma_{\min}(A) \leq \sigma_{\max}(A) \leq \sqrt{n} + \sqrt{m} + \delta. \tag{47}$$

The proof of this lemma can be found in Vershynin (2018).

# B  Non-existence of Global PL Constant and Smoothness Constant for Problem 2

In this section, we show that under mild assumptions, the PL inequality and smoothness inequality can only hold with constants $\mu_{\text{over}} = 0$ and $K_{\text{over}} = \infty$ for Problem 2.

We then make the following assumption on Problem 2.

**Assumption B.1.** $(W_1, W_2) = (0, 0)$ *is not a global minimizer of Problem 2.*

Based on the above assumption, one has the following proposition.

**Proposition B.1** (Non-existence of global PL constant and smoothness constant). *Under Assumption B.1, the PL inequality and smoothness inequality can only hold with constants $\mu_{over} = 0$ and $K_{over} = \infty$ for $L(W_1, W_2)$.*

*Proof.* We first show $\mu_{\text{over}} = 0$. The gradient of $L$ is given as follows

$$\|\nabla L(W_1, W_2)\|_F^2 = \|\nabla \ell(W) W_2\|_F^2 + \|\nabla \ell(W)^\top W_1\|_F^2. \tag{48}$$

Notice when $W_1, W_2$ are zero matrices, the RHS of the above equation is zero. Therefore, we have $\|\nabla L(W_1, W_2)\|_F^2 = 0$. On the other hand, under Assumption B.1, since $(W_1, W_2) = (0, 0)$ is not a minimizer of Problem 2, we have $L(W_1, W_2) \neq 0$. Thus, the PL inequality can only hold globally with $\mu_{\text{over}} = 0$,

$$\|\nabla L(W_1, W_2)\|_F^2 = \|\nabla \ell(W) W_2\|_F^2 + \|\nabla \ell(W)^\top W_1\|_F^2 \geq 2\mu_{\text{over}} L(W_1, W_2). \tag{49}$$

Then, we show $K_{\text{over}} = \infty$ for Problem 2. We consider the smoothness inequality evaluated on arbitrary $(W_1, W_2)$ and the minimizer $(W_1^*, W_2^*)$ of Problem 2:

$$L(W_1, W_2) \leq L(W_1^*, W_2^*) + \langle \nabla L(W_1^*, W_2^*), Z^* - Z \rangle + \frac{K_{\text{over}}}{2} \|Z^* - Z\|_F^2, \tag{50}$$

where we use $Z, Z^*$ in short for $(W_1, W_2), (W_1^*, W_2^*)$. Since $(W_1^*, W_2^*)$ minimizes Problem 2, we have $\nabla L(W_1^*, W_2^*) = 0_{(m+n) \times h}$ and $L(W_1^*, W_2^*) = 0$. Thus, equation 50 is equivalent to the following

$$K_{\text{over}} \geq \frac{2L(W_1, W_2)}{\|Z_W - Z_{W^*}\|_F^2} = \frac{2\ell(W_1 W_2^\top)}{\|Z_W - Z_{W^*}\|_F^2}. \tag{51}$$

On the other hand, since $\ell(W)$ is $\mu$-strongly convex w.r.t. $W$, the following inequality holds for arbitrary $U, V$

$$\ell(U) \geq \ell(V) + \langle \nabla \ell(V), U - V \rangle + \frac{\mu}{2} \|U - V\|_F^2. \tag{52}$$

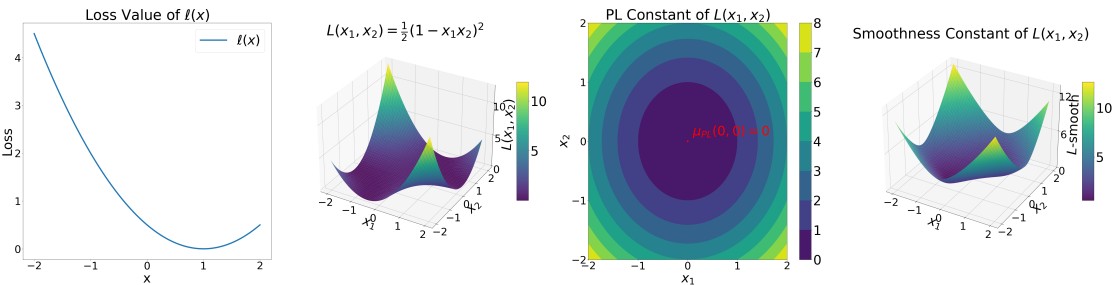

Figure 3: Plot of a toy example illustrating the loss function $\ell(x)$ and its overparametrized version $L(x_1, x_2)$, along with the corresponding local PL constant $\mu_{\text{over}}(x_1, x_2)$ and smoothness constant $K_{\text{over}}(x_1, x_2)$. The definitions of $\ell(x)$, $L(x_1, x_2)$, $\mu_{\text{over}}(x_1, x_2)$, and $K_{\text{over}}(x_1, x_2)$ are given in equation 57, equation 58, and equation 59.

We substitute $U, V$ with $W_1 W_2^\top, W_1^* (W_2^*)^\top$ in equation 52, we have

$$\ell(W_1 W_2^\top) \geq \frac{\mu}{2} \| W_1 W_2^\top - W_1^* (W_2^*)^\top \|_F^2 . \tag{53}$$

Finally, we combine equation 51 and equation 53, and derive the following lower bound on $K_{\text{over}}$

$$
\begin{aligned}
K_{\text{over}} &\geq \frac{2\ell(W_1 W_2^\top)}{\| Z_W - Z_{W^*} \|_F^2} && \text{Based on equation 51} \\
&\geq \frac{\mu \| W_1 W_2^\top - W_1^* (W_2^*)^\top \|_F^2}{\| W_1 - W_1^* \|_F^2 + \| W_2 - W_2^* \|_F^2} && \text{Apply equation 53 to } \ell(W_1 W_2^*) \\
&= \frac{\mu \| W_1 W_2^\top - W_1^* W_2^\top + W_1^* W_2^\top - W_1^* (W_2^*)^\top \|_F^2}{\| W_1 - W_1^* \|_F^2 + \| W_2 - W_2^* \|_F^2} \\
&= \frac{\mu \| (W_1 - W_1^*) W_2^\top + W_1^* (W_2 - W_2^*)^\top ) \|_F^2}{\| W_1 - W_1^* \|_F^2 + \| W_2 - W_2^* \|_F^2} \\
&\geq \frac{\mu}{2} \cdot \frac{\| (W_1 - W_1^*) W_2^\top \|_F^2 + \| W_1^* (W_2 - W_2^*)^\top ) \|_F^2}{2 \| W_1 - W_1^* \|_F^2 + \| W_2 - W_2^* \|_F^2} && \text{Apply Lemma A.1} \\
&\geq \frac{\mu}{2} \cdot \frac{\sigma_{\min}^2(W_2) \| W_1 - W_1^* \|_F^2 + \sigma_{\min}^2(W_1^*) \| W_2 - W_2^* \|_F^2}{\| W_1 - W_1^* \|_F^2 + \| W_2 - W_2^* \|_F^2} && \text{Apply Lemma A.1} . \tag{54}
\end{aligned}
$$

Similarly, one can also derive the following lower bound on $K_{\text{over}}$

$$K_{\text{over}} \geq \frac{\mu}{2} \cdot \frac{\sigma_{\min}^2(W_2^*) \| W_1 - W_1^* \|_F^2 + \sigma_{\min}^2(W_1) \| W_2 - W_2^* \|_F^2}{\| W_1 - W_1^* \|_F^2 + \| W_2 - W_2^* \|_F^2} \tag{55}$$

We take the average of the lower bound on $K_{\text{over}}$ in equation 54 and equation 55,

$$
\begin{aligned}
K_{\text{over}} &\geq \frac{\mu}{4} \cdot \frac{\left( \sigma_{\min}^2(W_2) + \sigma_{\min}^2(W_2^*) \right) \| W_1 - W_1^* \|_F^2 + \left( \sigma_{\min}^2(W_1) + \sigma_{\min}^2(W_1^*) \right) \| W_2 - W_2^* \|_F^2}{\| W_1 - W_1^* \|_F^2 + \| W_2 - W_2^* \|_F^2} \\
&\geq \frac{\mu}{4} \cdot \min \left( \sigma_{\min}^2(W_1) + \sigma_{\min}^2(W_1^*), \sigma_{\min}^2(W_2) + \sigma_{\min}^2(W_2^*) \right) \\
&\geq \frac{\mu}{4} \cdot \min \left( \sigma_{\min}^2(W_1), \sigma_{\min}^2(W_2) \right) . \tag{56}
\end{aligned}
$$

Due to the arbitrary choices of $W_1, W_2$, we can let $\sigma_{\min}(W_1)$ and $\sigma_{\min}(W_2)$ to be arbitrarily large, thus the smoothness inequality for Problem 2 can only hold globally with $K_{\text{over}} = \infty$.

To illustrate Proposition B.1, we examine how the PL constant and smoothness constant evolve in a simple one-dimensional setting. Specifically, we compare the *non-overparametrized* loss

$$\ell(x) = \frac{1}{2} \left( x - 1 \right)^2 \tag{57}$$

with the *overparametrized* loss

$$L(x_1, x_2) = \tfrac{1}{2} \left( x_1 x_2 - 1 \right)^2. \tag{58}$$

In this simple example, one can exactly compute local smoothness constants and PL constants as follows

$$\mu_{\text{over}}(x_1, x_2) = \frac{(\nabla L(x_1, x_2))^2}{2L(x_1, x_2)} = x_1^2 + x_2^2,$$

$$K_{\text{over}}(x_1, x_2) = \lambda_{\max}(\nabla^2 L(x_1, x_2)) = \frac{x_1^2 + x_2^2 + \sqrt{(x_1^2 + x_2^2)^2 - 4x_1^2 x_2^2 + (2x_1 x_2 - 1)^2}}{2}. \tag{59}$$

Since the local smoothness constants and PL constants of $L(x_1, x_2)$ now depend on the input, we use $\mu_{\text{over}}(x_1, x_2), K_{\text{over}}(x_1, x_2)$ to denote them.

In Figure 3, we plot the loss landscapes for both $\ell(x)$ and $L(x)$. We observe that while $\ell(x)$ is strongly convex and smooth, with global PL and smoothness constants $\mu = K = 1$, the overparameterized loss $L(x)$ is non-convex. Moreover, its local PL constant $\mu_{\text{over}}(x_1, x_2)$ vanishes when $x_1 = x_2 = 0$, and its local smoothness constant $K_{\text{over}}(x_1, x_2)$ diverges as $\|x_1\|^2 + \|x_2\|^2 \to \infty$. Consequently, if one attempts to enforce the PL inequality and Descent Lemma with *global* constants in the overparameterized setting, the only possibilities are

$$\mu_{\text{over}} = \min_{x_1, x_2} \mu_{\text{over}}(x_1, x_2) = 0 \quad \text{and} \quad K_{\text{over}} = \max_{x_1, x_2} K_{\text{over}}(x_1, x_2) = \infty. \tag{60}$$

$\square$

# C   A Detailed Comparison with Prior Work

In this section, we present a detailed comparison to Arora et al. (2018); Du et al. (2018a); Xu et al. (2023) to highlight the difference in technical details and improvement on the convergence rate.

**Summary of the strategy of proof in (Arora et al., 2018; Du et al., 2018a; Xu et al., 2023).** Based on GD update in equation 7, one can derive the following update on the product $W(t)$

$$W(t+1) = W(t) - \eta_t \mathcal{T}_t^* \circ \mathcal{T}_t(\nabla \ell(t)) + \eta_t^2 \nabla \ell(t) W(t)^\top \nabla \ell(t), \tag{61}$$

where $\mathcal{T}_t^*$ is the adjoint of $\mathcal{T}_t$. Then, substituting equation 61 into the smoothness inequality of the non-overparametrized model in equation 3, we can derive the following upper bound on the loss at iteration $t+1$ using the loss at iteration $t$ (Also see Lemma3.1 in Xu et al. (2023)).

**Lemma C.1.** *If at the $t$-th iteration of GD applied to the over-parametrized loss $L$, the step size $\eta_t$ satisfies*

$$\sigma_{\min}^2(\mathcal{T}_t) - \eta_t \|\nabla \ell(t)\|_F \|W(t)\|_F - \frac{K\eta_t}{2} \big[ \sigma_{\max}^2(\mathcal{T}_t) + \eta_t \|\nabla \ell(t)\|_F \|W(t)\|_F \big]^2 \geq 0, \tag{62}$$

*then the following inequality holds*

$$L(t+1) \leq \rho(\eta, t) L(t), \tag{63}$$

*where*

$$\rho(\eta, t) = 1 - 2\eta_t \mu \sigma_{\min}^2(\mathcal{T}_t) + K\mu \eta_t^2 \sigma_{\max}^4(\mathcal{T}_t) + 2\eta_t^2 \mu \sigma_{\max}(W(t)) \|\nabla \ell(t)\|_F$$
$$+ 2\eta_t^3 \mu K \sigma_{\max}^2(\mathcal{T}_t) \sigma_{\max}(W(t)) \|\nabla \ell(t)\|_F + \eta_t^4 \mu K \sigma_{\max}^2(W(t)) \|\nabla \ell(t)\|_F^2. \tag{64}$$

**Improvement of the local rate of decrease.** First, one can see the local rates of decrease in both work are polynomials of degree four and depend on $\eta_t, \nabla \ell(t)$ and singular values of $\mathcal{T}_t, W_t$. Moreover, around any global minimum, i.e., $L(t) \approx 0, \|\nabla \ell(t)\|_F \approx 0$, we have the following local rate of decrease per iteration

$$1 - 2\eta_t \mu \sigma_{\min}^2(\mathcal{T}_t) + \eta_t^2 K\mu \sigma_{\max}^4(\mathcal{T}_t) \qquad \text{local rate of decrease in prior work},$$
$$1 - 2\eta_t \mu \sigma_{\min}^2(\mathcal{T}_t) + \eta_t^2 K\mu \sigma_{\min}^2(\mathcal{T}_t) \eta_t^2 \sigma_{\max}^2(\mathcal{T}_t) \qquad \text{local rate of decrease in this work}, \tag{65}$$

and the optimal local rates of decrease regardless of the constraints on the $\eta_t$ are

$$1 - \frac{\mu}{K} \cdot \frac{\sigma_{\min}^4(\mathcal{T}_t)}{\sigma_{\max}^4(\mathcal{T}_t)} \qquad \text{optimal local rate of decrease in prior work},$$

$$1 - \frac{\mu}{K} \cdot \frac{\sigma_{\min}^2(\mathcal{T}_t)}{\sigma_{\max}^2(\mathcal{T}_t)} \qquad \text{optimal local rate of decrease in this work}. \tag{66}$$

Thus, one can see our characterization of local Descent lemma and PL inequality leads to faster local rates of decrease compared with prior results by $\frac{\sigma_{\min}^2(\mathcal{T}_t)}{\sigma_{\max}^2(\mathcal{T}_t)}$. Nevertheless, equation 66 does not imply linear convergence since if $\lim_{t \to \infty} \frac{\sigma_{\min}^2(\mathcal{T}_t)}{\sigma_{\max}^2(\mathcal{T}_t)} = 0$, one would not expect sufficient decrease per iteration. In order to show linear convergence, one needs to provide a uniform lower bound on $\frac{\sigma_{\min}^2(\mathcal{T}_t)}{\sigma_{\max}^2(\mathcal{T}_t)}, \forall t$.

**Improvement of the local rate of convergence.** In this work, we show when the step sizes satisfy certain constraints (See Theorem 3.2), there exist uniform spectral bounds for the condition number of $\mathcal{T}_t$, i.e., $\frac{\sigma_{\min}^2(\mathcal{T}_t)}{\sigma_{\max}^2(\mathcal{T}_t)} \leq c(\eta_0)\frac{\alpha_1}{\alpha_2}, \forall t$ where $\alpha_1, \alpha_2$ only depend on the initial weights and $c(\eta_0)$ is a constant approaching one as $\eta_0$ decreases. Thus, the optimal final rate of convergence derived in this work is

$$1 - \frac{\mu}{K} \cdot \frac{\alpha_1}{\alpha_2} \qquad \text{optimal local rate of convergence in this work}. \tag{67}$$

In prior work, the rates in (Du et al., 2018a; Arora et al., 2018) are extremely slow in practice (See Section 4 in (Xu et al., 2023)). In (Xu et al., 2023), the authors introduce two auxiliary constants $0 < c_1 < 1, c_2 > 1$, and show that one can uniformly bound the condition number of $\mathcal{T}_t$ during training, i.e., $\frac{\sigma_{\min}^2(\mathcal{T}_t)}{\sigma_{\max}^2(\mathcal{T}_t)} \leq \frac{c_1\alpha_1}{c_2\alpha_2}, \forall t$. Moreover, they enforce problem-dependent assumptions on the choices of $c_1, c_2$. According to Claim E.1 in (Xu et al., 2023), $\frac{c_1}{c_2}$ is at most $\frac{1}{3}$ and can be arbitrarily small when the initial loss is large. Thus, the local rate of convergence in (Xu et al., 2023) is at most, in our notation,

$$1 - \frac{\mu}{K} \cdot \frac{c_1^2 \alpha_1^2}{c_2^2 \alpha_2^2} \qquad \text{optimal local rate of convergence in (Xu et al., 2023)}. \tag{68}$$

When comparing equation 67 and equation 68, one can directly conclude the local rate of convergence derived in this work is much faster than the rate derived in (Xu et al., 2023). Moreover, the optimal local rate of convergence of the overparametrized model in this work is different from the optimal rate of convergence of the non-overparametrized model up to a factor of $\frac{\alpha_1}{\alpha_2}$, which shows overparametrization has a benign effect if one can control $\frac{\alpha_1}{\alpha_2}$ through properly initialization of the weights. However, such results are not shown in the work of (Arora et al., 2018; Du et al., 2018a; Xu et al., 2023).

## D   Proof of Theorem 3.1

In this section, we present the proof of Theorem 3.1.

**Theorem D.1** (Restate of Theorem 3.1)**.** *At the $t$-th iteration of GD applied to the Problem 2, the Descent lemma and PL inequality hold with local smoothness constant $K_t$ and PL constant $\mu_t$, i.e.,*

$$L(t+1) \leq L(t) - \left(\eta_t - \frac{K_t \eta_t^2}{2}\right)\|\nabla L(t)\|_F^2, \quad \frac{1}{2}\|\nabla L(t)\|_F^2 \geq \mu_t L(t). \tag{69}$$

*Moreover, if the step size $\eta_t$ satisfies $\eta_t > 0$ and $\eta_t K_t < 2$, then the following inequality holds*

$$L(t+1) \leq L(t)(1 - 2\mu_t \eta_t + \mu_t K_t \eta_t^2) := L(t)\rho(\eta_t, t), \tag{70}$$

*where*

$$\mu_t = \mu\sigma_{\min}^2(\mathcal{T}_t), \tag{71}$$

$$K_t = K\sigma_{\max}^2(\mathcal{T}_t) + \sqrt{2KL(t)} + 6K^2\sigma_{\max}(W(t))L(t)\eta_t^2 + 3K\sigma_{\max}^2(\mathcal{T}_t)\sqrt{2KL(t)}\eta_t, \tag{72}$$

*and we use $L(t), \mathcal{T}_t$ as a shorthand for $L(W_1(t), W_2(t)), \mathcal{T}(\,\cdot\,; W_1(t), W_2(t))$ resp.*

*Proof.* We first show that the local PL inequality holds.

$$
\begin{aligned}
\|\nabla L(t)\|_F^2 &= \left\| \begin{bmatrix} \nabla \ell(t) W_2(t) \\ \nabla \ell(t)^\top W_1(t) \end{bmatrix} \right\|_F^2 \\
&= \|\ell(t) W_2(t)\|_F^2 + \|\ell(t)^\top W_1(t)\|_F^2 \\
&\geq \sigma_{\min}^2(W_2(t)) \|\nabla \ell(t)\|_F^2 + \sigma_{\min}^2(W_1(t)) \|\nabla \ell(t)\|_F^2 \qquad &&\text{Apply Lemma A.1} \\
&\geq 2\mu \sigma_{\min}^2(W_2(t)) L(t) + 2\mu \sigma_{\min}^2(W_1(t)) L(t) \qquad &&\text{Apply PL inequality of } \ell \\
&= 2\mu_t L(t) \,,
\end{aligned}
$$

where the last equality uses the fact that $\sigma_{\min}^2(\mathcal{T}_t) = \sigma_{\min}^2(W_1(t)) + \sigma_{\min}^2(W_2(t))$. Then, we show that Descent lemma holds with local smoothness constant $K_t$. We can view $L(t+1)$ using the following second Taylor approximation,

$$
\begin{aligned}
L(t+1) &= L(t) + \langle \nabla L(t), Z_{t+1} - Z_t \rangle + \int_0^1 (1-\tau) \langle Z_{t+1} - Z_t, H(\tau)(Z_{t+1} - Z_t) \rangle d\tau \,, \\
&= L(t) - \eta_t \|\nabla L(t)\|_F^2 + \eta_t^2 \|\nabla L(t)\|_F^2 \int_0^1 (1-\tau)\langle g_t, H(\tau) g_t \rangle d\tau \,, \qquad (73)
\end{aligned}
$$

where we use $Z_{t+1}, Z_t$ in short for $(W_1(t+1), W_2(t+1)), (W_1(t), W_2(t))$ respectively, and $g_t = \frac{\nabla L(t)}{\|\nabla L(t)\|_F}$ to denote the unit vector of the gradient direction. Moreover, the $H(\tau)$ is defined as follows,

$$
\begin{aligned}
H(\tau) &= \nabla^2 L\big((1-\tau)W_1(t) + \tau W_1(t+1), (1-\tau)W_2(t) + \tau W_2(t+1)\big) \\
&= \nabla^2 L\big(W_1(t) - \eta_t \tau \nabla_{W_1} L(t), W_2(t) - \eta_t \tau \nabla_{W_2} L(t)\big) \,. \qquad (74)
\end{aligned}
$$

Notice the integral in the equation 73 does not have a closed-form solution. We use the following two-step approach to derive an upper bound on the RHS of equation 73.

**Step one.** We first show that one can upper bound $\langle g_t, H(0) g_t \rangle$ using the singular values of $\mathcal{T}_t$, $K$ and $L(t)$. The following lemma characterizes it formally.

**Lemma D.1** (Upper bound on $\langle g_t, H(0) g_t \rangle$). *We have the following upper bound*

$$
\langle g_t, H(0) g_t \rangle \leq K \sigma_{\max}^2(\mathcal{T}_t) + \sqrt{2KL(t)} \,. \qquad (75)
$$

The proof of Lemma D.1 is presented at the end of this section.

**Step two.** Then, for any $\tau \in [0, 1)$, we can show $|\langle g_t, \big(H(0) - H(\tau)\big) g_t \rangle|$ is bounded, which leads to an upper bound on $\langle g_t, H(\tau) g_t \rangle$. The following lemma characterizes the upper bound on $\langle g_t, H(\tau) g_t \rangle$.

**Lemma D.2** (Uniform upper bound on $\langle g_t, H(\tau) g_t \rangle$). *For any $\tau \in [0, 1)$, we have*

$$
|\langle g_t, H(\tau) g_t \rangle| \leq K \sigma_{\max}^2(\mathcal{T}_t) + \sqrt{2KL(t)} + 6K^2 \sigma_{\max}(W(t)) L(t) \eta_t^2 + 3K \sigma_{\max}^2(\mathcal{T}_t) \sqrt{2KL(t)} \eta_t := K_t \,.
$$

The proof of Lemma D.2 is presented at the end of this section.

Based on equation 73 and Lemma D.2, one can derive Descent lemma

$$L(t+1) = L(t) - \eta_t \|\nabla L(t)\|_F^2 + \eta_t^2 \|\nabla L(t)\|_F^2 \int_0^1 (1-\tau)\langle g_t, H(\tau)g_t\rangle d\tau \qquad \text{Equation 73}$$

$$\leq L(t) - \eta_t \|\nabla L(t)\|_F^2 + \eta_t^2 \|\nabla L(t)\|_F^2 \int_0^1 (1-\tau)\max_\tau |\langle g_t, H(\tau)g_t\rangle| d\tau$$

$$\leq L(t) - \eta_t \|\nabla L(t)\|_F^2 + \eta_t^2 \|\nabla L(t)\|_F^2 \int_0^1 (1-\tau)K_t d\tau \qquad \text{Lemma D.2}$$

$$= L(t) - \eta_t \|\nabla L(t)\|_F^2 + \frac{\eta_t^2 K_t}{2}\|\nabla L(t)\|_F^2$$

$$= L(t) - (\eta_t - \frac{\eta_t^2 K_t}{2})\|\nabla L(t)\|_F^2. \tag{76}$$

Therefore, Descent lemma is proved. $\qquad\square$

Now we present the proof of Lemma D.1 and Lemma D.2. We first define the following quantity which will be used in the proof

$$M(s) = L(W_1(t) - s\eta_t \nabla_{W_1}L(t), W_2(t) - s\eta_t \nabla_{W_2}L(t)), \tag{77}$$

$$A(s) = W(t) - s\eta_t\big(\nabla_{W_2}L(t)W_2(t)^\top + W_1(t)\nabla_{W_1}L(t)^\top\big) + s^2\eta_t^2\nabla_{W_1}L(t)\nabla_{W_2}L(t)^\top, \tag{78}$$

where $A(s)$ is the product of $W_1(t) - s\eta_t\nabla_{W_1}L(t)$ and $W_2(t) - s\eta_t\nabla_{W_2}L(t)$. Moreover, we have $M(0) = L(t)$, $M(\eta_t) = L(t+1)$ and $M(s) = \ell(A(s))$.

Then, we present several lemmas that will be used in the proof of Lemma D.1 and Lemma D.2.

**Lemma D.3.** *Given $W_1(t) \in \mathbb{R}^{n\times h}, W_2(t) \in \mathbb{R}^{m\times h}$ at $t$-th iteration, the following holds*

$$2\|\nabla_{W_1}L(t)\nabla_{W_2}L(t)^\top\|_F \leq \|\nabla_{W_1}L(t)\|_F^2 + \|\nabla_{W_2}L(t)\|_F^2 \tag{79}$$

$$\|\nabla_{W_1}L(t)\nabla_{W_2}L(t)^\top\|_F \leq 2K\sigma_{\max}(W(t))L(t). \tag{80}$$

*Proof.* Based on Lemma A.1, one has $2\|AB\|_F \leq \|A\|_F^2 + \|B\|_F^2$. Thus, let $A = \nabla_{W_1}L(t), B = \nabla_{W_2}L(t)$, and we complete the proof of equation 79.

For equation 80, one has the following

$$\|\nabla_{W_1}L(t)\nabla_{W_2}L(t)^\top\|_F = \|\nabla\ell(t)W(t)^\top\nabla\ell(t)^\top\|_F$$

$$\leq \sigma_{\max}(W(t))\|\nabla\ell(t)\|_F^2 \qquad \text{equation 37 in Lemma A.1}$$

$$\leq 2K\sigma_{\max}(W(t))L(t) \qquad K\text{-smooth of } \ell, \tag{81}$$

which completes the proof. $\qquad\square$

**Lemma D.4.** *Given $W_1(t) \in \mathbb{R}^{n\times h}, W_2(t) \in \mathbb{R}^{m\times h}$ at $t$-th iteration, the following holds*

$$\|\nabla_{W_2}L(t)W_2(t)^\top + W_1(t)\nabla_{W_1}L(t)^\top\|_F \leq \sigma_{\max}^2(\mathcal{T}_t)\sqrt{2KL(t)}. \tag{82}$$

*Proof.* We prove this lemma using the results from Lemma A.1 and Lemma A.2

$$\|\nabla_{W_2}L(t)W_2(t)^\top + W_1(t)\nabla_{W_1}L(t)^\top\|_F$$

$$\leq \|\nabla_{W_2}L(t)W_2(t)^\top\|_F + \|W_1(t)\nabla_{W_1}L(t)^\top\|_F \qquad \text{Property of norm}$$

$$= \|\nabla\ell(t)W_2W_2(t)^\top\|_F + \|W_1(t)W_1(t)^\top\nabla\ell(t)^\top\|_F \qquad \text{See definition of } \mathcal{T}_t$$

$$\leq \sigma_{\max}^2(W_2(t))\|\nabla\ell(t)\|_F^2 + \sigma_{\max}^2(W_1(t))\|\nabla\ell(t)\|_F^2 \qquad \text{equation 37 in Lemma A.1}$$

$$= \sigma_{\max}^2(\mathcal{T}_t)\|\nabla\ell(t)\|_F^2 \qquad \text{Lemma A.2}$$

$$\leq \sigma_{\max}^2(\mathcal{T}_t)\sqrt{2KL(t)} \qquad K\text{-smooth of } \ell. \tag{83}$$

$\qquad\square$

**Lemma D.5.** *Given* $W_1(t) \in \mathbb{R}^{n \times h}, W_2(t) \in \mathbb{R}^{m \times h}$ *at* $t$*-th iteration, for any* $s \in (0, 1]$*, the following holds*

$$\|\nabla\ell\big(A(s)\big) - \nabla\ell\big(A(0)\big)\|_F \leq \eta_t K \sqrt{2KL(t)}\sigma^2_{\max}(\mathcal{T}_t) + 2\eta_t^2 K^2 \sigma_{\max}(W(t))L(t) \tag{84}$$

*Proof.* Based on Lemma A.1 and the assumption that $\ell$ is $K$-smooth, one has the following

$$
\begin{aligned}
&\|\nabla\ell\big(A(s)\big) - \nabla\ell\big(A(0)\big)\|_F \\
\leq & K\|A(s) - A(0)\|_F \\
= & K\| - s\eta_t \big(\nabla_{W_2}L(t)W_2(t)^\top + W_1(t)\nabla_{W_1}L(t)^\top\big) + s^2\eta_t^2\nabla_{W_1}L(t)\nabla_{W_2}L(t)^\top\|_F \\
\leq & s\eta_t K\|\nabla_{W_2}L(t)W_2(t)^\top + W_1(t)\nabla_{W_1}L(t)^\top\|_F + s^2\eta_t^2 K\|\nabla_{W_1}L(t)\nabla_{W_2}L(t)^\top\|_F \\
\leq & \eta_t K\|\nabla_{W_2}L(t)W_2(t)^\top + W_1(t)\nabla_{W_1}L(t)^\top\|_F + \eta_t^2 K\|\nabla_{W_1}L(t)\nabla_{W_2}L(t)^\top\|_F \,,
\end{aligned}
\tag{85}
$$

where the last line is due to the fact that $s \in (0, 1]$.

Then, based on Lemma D.3 and Lemma D.4, one has the following,

$$
\begin{aligned}
&\|\nabla\ell\big(A(s)\big) - \nabla\ell\big(A(0)\big)\|_F \\
\leq & K\|A(s) - A(0)\|_F \\
\leq & \eta_t K\|\nabla_{W_2}L(t)W_2(t)^\top + W_1(t)\nabla_{W_1}L(t)^\top\|_F + \eta_t^2 K\|\nabla_{W_1}L(t)\nabla_{W_2}L(t)^\top\|_F \\
\leq & \eta_t K\sigma^2_{\max}(\mathcal{T}_t)\sqrt{2KL(t)} + \eta_t^2 K \cdot 2K\sigma_{\max}(W(t))L(t) \,,
\end{aligned}
\tag{86}
$$

which completes the proof. □

**Lemma D.1** (Upper bound on $\langle g_t, H(0)g_t \rangle$)**.** *We have the following upper bound*

$$\langle g_t, H(0)g_t \rangle \leq K\sigma^2_{\max}(\mathcal{T}_t) + \sqrt{2KL(t)} \,. \tag{87}$$

*Proof.* First, we notice $\langle g_t, H(0)g_t \rangle$ is the second-order directional derivative of $L(t)$ w.r.t. the gradient direction,

$$\langle g_t, H(0)g_t \rangle = \frac{1}{\|\nabla L(t)\|_F^2} \cdot \frac{d^2}{ds^2}M(s)\Big|_{s=0} \,. \tag{88}$$

Moreover, we can compute $\frac{d^2}{ds^2}M(s)\Big|_{s=0}$ as follows

$$
\begin{aligned}
&\frac{d^2}{ds^2}M(s)\Big|_{s=0} \\
= & \frac{d^2}{ds^2}L\bigg(\big(W_1(t) - s\nabla_{W_1}L(t)\big)\big(W_2(t) - s\nabla_{W_2}L(t)\big)^\top\bigg)\Big|_{s=0} \\
= & \frac{d^2}{ds^2}\ell\big(A(s)\big)\Big|_{s=0} & \text{Definition of } A(s) \\
= & \frac{d}{ds}\big\langle \nabla\ell\big(A(s)\big), \frac{d}{ds}A(s)\big\rangle\Big|_{s=0} \\
= & \big\langle \nabla\ell\big(A(s)\big), \frac{d^2}{ds^2}A(s)\big\rangle + \big\langle \frac{d}{ds}A(s), \nabla^2\ell\big(A(s)\big)\frac{d}{ds}A(s)\big\rangle\Big|_{s=0} \,.
\end{aligned}
\tag{89}
$$

Under the assumption that $\ell$ is $K$-smooth and Lemma A.1, one can derive the following upper bound on $\frac{d^2}{ds^2}M(s)\big|_{s=0}$

$$
\begin{aligned}
&\frac{d^2}{ds^2}M(s)\bigg|_{s=0}\\
&= \left\langle \nabla\ell\big(A(s)\big), \frac{d^2}{ds^2}A(s)\right\rangle + \left\langle \frac{d}{ds}A(s), \nabla^2\ell(A(s))\frac{d}{ds}A(s)\right\rangle\bigg|_{s=0}\\
&\leq \left\langle \nabla\ell\big(A(s)\big), \frac{d^2}{ds^2}A(s)\right\rangle + K\|\frac{d}{ds}A(s)\|_F^2\bigg|_{s=0} && \ell \text{ is } K\text{-smooth}\\
&= 2\langle \nabla\ell(t), \nabla_{W_1}L(t)\nabla_{W_2}L(t)^\top\rangle + K\|\nabla_{W_2}L(t)W_2(t)^\top + W_1(t)\nabla_{W_1}L(t)^\top\|_F^2\\
&\leq 2\|\nabla\ell(t)\|_F \cdot \|\nabla_{W_1}L(t)\nabla_{W_2}L(t)^\top\|_F\\
&\quad + K[\sigma_{\max}^2(W_1(t)) + \sigma_{\max}^2(W_2(t))]\cdot[\|\nabla_{W_1}L(t)\|_F^2 + \|\nabla_{W_2}L(t)\|_F^2] && \text{Lemma A.1}\\
&\leq \|\nabla\ell(t)\|_F \cdot [\|\nabla_{W_1}L(t)\|_F^2 + \|\nabla_{W_2}L(t)\|_F^2]\\
&\quad + K[\sigma_{\max}^2(W_1(t)) + \sigma_{\max}^2(W_2(t))]\cdot[\|\nabla_{W_1}L(t)\|_F^2 + \|\nabla_{W_2}L(t)\|_F^2] && \text{Lemma A.1}\\
&\leq \sqrt{2KL(t)}\cdot[\|\nabla_{W_1}L(t)\|_F^2 + \|\nabla_{W_2}L(t)\|_F^2] && K\text{-smooth of } \ell\\
&\quad + K[\sigma_{\max}^2(W_1(t)) + \sigma_{\max}^2(W_2(t))]\cdot[\|\nabla_{W_1}L(t)\|_F^2 + \|\nabla_{W_2}L(t)\|_F^2]. && (90)
\end{aligned}
$$

Finally, we derive the the upper bound on $\langle g_t, H(0)g_t\rangle$ based on equation 90

$$
\begin{aligned}
\langle g_t, H(0)g_t\rangle &= \frac{1}{\|\nabla L(t)\|_F^2}\cdot\frac{d^2}{ds^2}M(s)\bigg|_{s=0}\\
&\leq \frac{[\|\nabla_{W_1}L(t)\|_F^2 + \|\nabla_{W_2}L(t)\|_F^2]\cdot\big(\sqrt{2KL(t)} + \sigma_{\max}^2(W_1(t)) + \sigma_{\max}^2(W_2(t))\big)}{\|\nabla_{W_1}L(t)\|_F^2 + \|\nabla_{W_2}L(t)\|_F^2}\\
&= \sqrt{2KL(t)} + \sigma_{\max}^2(W_1(t)) + \sigma_{\max}^2(W_2(t))\\
&= \sqrt{2KL(t)} + \sigma_{\max}^2(\mathcal{T}_t), && (91)
\end{aligned}
$$

where the last line is based on Lemma A.2. $\qquad\square$

**Lemma D.2** (Upper bound on $\langle g_t, H(\tau)g_t\rangle$). *For any $\tau \in [0,1)$, we have*

$$
\langle g_t, H(\tau)g_t\rangle \leq K_t, \tag{92}
$$

*where*

$$
K_t = K\sigma_{\max}^2(\mathcal{T}_t) + \sqrt{2KL(t)} + 6K^2\sigma_{\max}^2(W(t))L(t)\eta_t^2 + 3K\sigma_{\max}^2(\mathcal{T}_t)\sqrt{2KL(t)}\eta_t. \tag{93}
$$

*Proof.* First, we use the same method to compute $\langle g_t, H(\tau)g_t\rangle$ as it was done in Lemma D.1

$$
\langle g_t, H(\tau)g_t\rangle = \frac{1}{\|\nabla L(t)\|_F^2}\cdot\frac{d^2}{ds^2}M(s+\tau)\bigg|_{s=0}. \tag{94}
$$

Based on similar calculations in equation 89, one has

$$
\begin{aligned}
&\frac{d^2}{ds^2}M(s+\tau)\bigg|_{s=0}\\
&= \frac{d^2}{ds^2}\ell\big(A(s+\tau)\big)\bigg|_{s=0}\\
&= \frac{d}{ds}\left\langle \nabla\ell\big(A(s+\tau)\big), \frac{d}{ds}A(s+\tau)\right\rangle\bigg|_{s=0}\\
&= \left\langle \nabla\ell\big(A(s+\tau)\big), \frac{d^2}{ds^2}A(s+\tau)\right\rangle + \left\langle \frac{d}{ds}A(s+\tau), \nabla^2\ell(A(s+\tau))\frac{d}{ds}A(s+\tau)\right\rangle\bigg|_{s=0}. && (95)
\end{aligned}
$$

Under the assumption that $\ell$ is $K$-smooth, Lemma A.1 and Lemma D.1, one can show

$$
\begin{aligned}
&\frac{d^2}{ds^2}M(s+\tau)\Big|_{s=0} \\
&= \Big\langle\nabla\ell\big(A(s+\tau)\big), \frac{d^2}{ds^2}A(s+\tau)\Big\rangle + \Big\langle\frac{d}{ds}A(s+\tau), \nabla^2\ell(A(s+\tau))\frac{d}{ds}A(s+\tau)\Big\rangle\Big|_{s=0} \\
&\leq \Big\langle\nabla\ell\big(A(s+\tau)\big), \frac{d^2}{ds^2}A(s+\tau)\Big\rangle + K\|\frac{d}{ds}A(s+\tau)\|_F^2\Big|_{s=0} \\
&= 2\big\langle\nabla\ell\big(A(\tau)\big), \nabla_{W_1}L(t)\nabla_{W_2}L(t)^\top\big\rangle \\
&\quad + K\|\nabla_{W_2}L(t)W_2(t)^\top + W_1(t)\nabla_{W_1}L(t)^\top - 2\tau\eta_t\nabla_{W_1}L(t)\nabla_{W_2}L(t)^\top\|_F^2 \\
&= 2\big\langle\nabla\ell\big(A(\tau)\big)-\nabla\ell\big(A(0)\big), \nabla_{W_1}L(t)\nabla_{W_2}L(t)^\top\big\rangle + 2\big\langle\nabla\ell\big(A(0)\big), \nabla_{W_1}L(t)\nabla_{W_2}L(t)^\top\big\rangle \\
&\quad + K\|\nabla_{W_2}L(t)W_2(t)^\top + W_1(t)\nabla_{W_1}L(t)^\top\|_F^2 + 4\tau^2\eta_t^2 K\|\nabla_{W_1}L(t)\nabla_{W_2}L(t)^\top\|_F^2 \\
&\quad - 4\tau K\eta_t\big\langle\nabla_{W_2}L(t)W_2(t)^\top + W_1(t)\nabla_{W_1}L(t)^\top, \nabla_{W_1}L(t)\nabla_{W_2}L(t)^\top\big\rangle \\
&\leq 2\big\langle\nabla\ell\big(A(0)\big), \nabla_{W_1}L(t)\nabla_{W_2}L(t)^\top\big\rangle + K\|\nabla_{W_2}L(t)W_2(t)^\top + W_1(t)\nabla_{W_1}L(t)^\top\|_F^2 \\
&\quad + 2\|\nabla\ell\big(A(\tau)\big)-\nabla\ell\big(A(0)\|_F \cdot \|\nabla_{W_1}L(t)\nabla_{W_2}L(t)^\top\|_F \\
&\quad + 4\tau^2\eta_t^2 K\|\nabla_{W_1}L(t)\nabla_{W_2}L(t)^\top\|_F^2 \\
&\quad + 4\tau\eta_t K\|\nabla_{W_2}L(t)W_2(t)^\top + W_1(t)\nabla_{W_1}L(t)^\top\|_F \cdot \|\nabla_{W_1}L(t)\nabla_{W_2}L(t)^\top\|_F \\
&\leq 2\big\langle\nabla\ell\big(A(0)\big), \nabla_{W_1}L(t)\nabla_{W_2}L(t)^\top\big\rangle + K\|\nabla_{W_2}L(t)W_2(t)^\top + W_1(t)\nabla_{W_1}L(t)^\top\|_F^2 \\
&\quad + 2\|\nabla\ell\big(A(\tau)\big)-\nabla\ell\big(A(0)\|_F \cdot \|\nabla_{W_1}L(t)\nabla_{W_2}L(t)^\top\|_F \\
&\quad + 4\eta_t^2 K\|\nabla_{W_1}L(t)\nabla_{W_2}L(t)^\top\|_F^2 \\
&\quad + 4\eta_t K\|\nabla_{W_2}L(t)W_2(t)^\top + W_1(t)\nabla_{W_1}L(t)^\top\|_F \cdot \|\nabla_{W_1}L(t)\nabla_{W_2}L(t)^\top\|_F \,,
\end{aligned}
\tag{96}
$$

where the last line is derived based on the fact that $\tau \in (0,1]$.

Notice in equation 90, we have shown

$$
\begin{aligned}
2\big\langle\nabla\ell\big(A(0)\big), \nabla_{W_1}L(t)\nabla_{W_2}L(t)^\top\big\rangle + K\|\nabla_{W_2}L(t)W_2(t)^\top + W_1(t)\nabla_{W_1}L(t)^\top\|_F^2 \\
\leq [\|\nabla_{W_1}L(t)\|_F^2 + \|\nabla_{W_2}L(t)\|_F^2]\cdot\big(\sqrt{2KL(t)} + \sigma_{\max}^2(\mathcal{T}_t)\big)\,.
\end{aligned}
\tag{97}
$$

Moreover, in Lemma D.3, Lemma D.4 and Lemma D.5, we have shown

$$
\begin{aligned}
2\|\nabla_{W_1}L(t)\nabla_{W_2}L(t)^\top\|_F &\leq \|\nabla_{W_1}L(t)\|_F^2 + \|\nabla_{W_2}L(t)\|_F^2 \\
\|\nabla_{W_1}L(t)\nabla_{W_2}L(t)^\top\|_F &\leq 2K\sigma_{\max}(W(t))L(t) \\
\|\nabla_{W_2}L(t)W_2(t)^\top + W_1(t)\nabla_{W_1}L(t)^\top\|_F &\leq \sigma_{\max}^2(\mathcal{T}_t)\sqrt{2KL(t)} \\
\|\nabla\ell\big(A(s)\big)-\nabla\ell\big(A(0)\|_F &\leq \eta_t K\sqrt{2KL(t)}\sigma_{\max}^2(\mathcal{T}_t) + 2\eta_t^2 K^2\sigma_{\max}(W(t))L(t)\,.
\end{aligned}
\tag{98,99}
$$

Thus, one can further upper bound equation 96 as follows

$$
\begin{aligned}
\frac{d^2}{ds^2} & M(s+\tau)\Big|_{s=0} \\
\leq & 2\big\langle \nabla\ell\big(A(0)\big), \nabla_{W_1}L(t)\nabla_{W_2}L(t)^\top\big\rangle + K\|\nabla_{W_2}L(t)W_2(t)^\top + W_1(t)\nabla_{W_1}L(t)^\top\|_F^2 \\
& + 2\|\nabla\ell\big(A(\tau)\big) - \nabla\ell\big(A(0)\|_F \cdot \|\nabla_{W_1}L(t)\nabla_{W_2}L(t)^\top\|_F \\
& + 4\eta_t^2 K\|\nabla_{W_1}L(t)\nabla_{W_2}L(t)^\top\|_F^2 \\
& + 4\eta_t K\|\nabla_{W_2}L(t)W_2(t)^\top + W_1(t)\nabla_{W_1}L(t)^\top\|_F \cdot \|\nabla_{W_1}L(t)\nabla_{W_2}L(t)^\top\|_F \\
\leq & [\|\nabla_{W_1}L(t)\|_F^2 + \|\nabla_{W_2}L(t)\|_F^2] \cdot \big(\sqrt{2KL(t)} + \sigma_{\max}^2(\mathcal{T}_t)\big) \\
& + \Big(\eta_t K\sqrt{2KL(t)}\sigma_{\max}^2(\mathcal{T}_t) + 2\eta_t^2 K^2 \sigma_{\max}(W(t))L(t)\Big) \cdot [\|\nabla_{W_1}L(t)\|_F^2 + \|\nabla_{W_2}L(t)\|_F^2] \\
& + 4\eta_t^2 K^2 \sigma_{\max}(W(t))L(t) \cdot [\|\nabla_{W_1}L(t)\|_F^2 + \|\nabla_{W_2}L(t)\|_F^2] \\
& + 2\eta_t K\sigma_{\max}^2(\mathcal{T}_t)\sqrt{2KL(t)} \cdot [\|\nabla_{W_1}L(t)\|_F^2 + \|\nabla_{W_2}L(t)\|_F^2].
\end{aligned}
\tag{100}
$$

As a result, we can show

$$
\begin{aligned}
\langle g_t, H(\tau)g_t\rangle &= \frac{1}{\|\nabla L(t)\|_F^2} \cdot \frac{d^2}{ds^2}M(s+\tau)\Big|_{s=0} \\
&\leq K\sigma_{\max}^2(\mathcal{T}_t) + \sqrt{2KL(t)} + 6K^2\sigma_{\max}(W(t))L(t)\eta_t^2 + 3K\sigma_{\max}^2(\mathcal{T}_t)\sqrt{2KL(t)}\eta_t.
\end{aligned}
$$

$\square$

# E  Proof of Theorem 3.2

In this section, we first introduce the generalized form of Theorem 3.2. Then, we provide a detailed proof.

**Theorem E.1** (Linear convergence of GD for Problem 2). *Assume the GD algorithm equation 7 is initialized such that $\alpha_1 > 0$. Pick any $0 < c < 1, d > 1$. Let $\eta_0^{(1)}$ be the unique positive solution of the following equation*

$$
\eta_0\big(\sqrt{2KL(0)} + 6K^2\beta_2 L(0)\eta_0^2 + K\exp(\sqrt{\eta_0})\alpha_2[1 + 3\sqrt{2KL(0)}\eta_0]\big) = 1,
\tag{101}
$$

*and $\eta_0^{(2)}$ be the smallest positive solution of the following equation[5]*

$$
4KL(0)\eta_0^2 = (1 - \exp(-\eta_0^c)) \times (1 - \Delta).
\tag{102}
$$

*Then, we define $\eta_{\max} = \min(\eta_0^{(1)}, \eta_0^{(2)}, \log\big(1 + \frac{\alpha_1}{2\alpha_2}\big)^{\frac{1}{c}})$. For any $\eta_0$ and $\eta_t$ such that $0 < \eta_0 \leq \eta_{\max}$ and $\eta_t$ satisfies*

$$
\eta_0 \leq \eta_t \leq \min\big((1 + \eta_0^d)^{\frac{t}{2}}\eta_0, \frac{1}{K_t}\big),
\tag{103}
$$

*one can derive the following linear convergence rate for GD*

$$
L(t+1) \leq L(t)\bar{\rho}(\eta_t, t) \leq L(t)\bar{\rho}(\eta_0, 0) \leq L(0)\bar{\rho}(\eta_0, 0)^{t+1},
\tag{104}
$$

*where*

$$
\begin{aligned}
\bar{\rho}(\eta_t, t) &= 1 - 2\bar{\mu}\eta_t + \bar{\mu}\bar{K}_t\eta_t^2, \quad \bar{\mu} = \mu\big[\alpha_1 + 2\alpha_2\big(1 - \exp(\eta_0^c)\big)\big], \quad \Delta = (1 + \eta_0^d)\bar{\rho}(\eta_0, 0), \\
\bar{K}_t &= \sqrt{2KL(0)\bar{\rho}(\eta_0, 0)^t} + 6K^2\beta_2 L(0)\eta_0^2\Delta^t + K\exp(\sqrt{\eta_0})\alpha_2\big[1 + 3\sqrt{2KL(0)\Delta^t}\eta_0\big].
\end{aligned}
$$

---

[5]In the case when equation 102 does not have positive solution, we set $\eta_0^{(2)} = \infty$.

Notice Theorem 3.2 in §3.2 can be viewed as a special case of Theorem E.1 with $c = \frac{1}{2}, d = 2$.

Before presenting the proof Theorem E.1, we first show that the constraints on $\eta_0$ do not induce an empty set, or equivalently $\eta_{\max} > 0$. Alongside, we provide several inequalities that are implied by the constraints on $\eta_0$, which the proof Theorem E.1 is relied on.

**Existence of $\eta_0$ .** To show the existence of $\eta_0$, it is equivalent to show that $\eta_{\max} > 0$. First, since $\alpha_1, \alpha_2$ are positive, we have $\log\left(1 + \frac{\alpha_1}{2\alpha_2}\right)^{\frac{1}{c}} > 0$. Moreover, one can see when $\eta_0 < \log\left(1 + \frac{\alpha_1}{2\alpha_2}\right)^{\frac{1}{c}}$, we have $\bar{\mu} > 0$.

Second, we show $\eta_0^{(1)} > 0$. The LHS of equation 101 increases as $\eta_0$ increases, and it equals zero as $\eta_0 = 0$. Thus, there exists a unique positive solution of equation 101, which is equivalent to $\eta_0^{(1)} > 0$. Notice

$$\bar{K}_0 = \sqrt{2KL(0)} + 6K^2\beta_2 L(0)\eta_0^2 + K\exp(\sqrt{\eta_0})\alpha_2\left[1 + 3\sqrt{2KL(0)}\eta_0\right]. \tag{105}$$

Therefore, $0 < \eta_0 \leq \eta_0^{(1)}$ implies $0 < \eta_0\bar{K}_0 \leq 1$ which is equivalent to $\eta_0 \leq \frac{1}{\bar{K}_0}$. This constraint further leads to $0 < \bar{\rho}(\eta_0, 0) < 1$ and $\Delta > 0$.

Finally, we show $\eta_0^{(2)} > 0$. Notice when $\eta_0 = 0$, the RHS and LHS of equation 102 both equal zero. Moreover, when $\eta_0 > 0$, one can rewrite equation 102 as follows

$$4KL(0)\eta_0^2 = (1 - \exp(-\eta_0^c)) \times (1 - \Delta)$$
$$\iff 4KL(0)\eta_0^2 = (1 - \exp(-\eta_0^c)) \times (2\bar{\mu}\eta_0 - \bar{\mu}\bar{K}_0\eta_0^2 - \eta_0^d\bar{\rho}(\eta_0, 0))$$
$$\iff 4KL(0)\eta_0^{1-c} = \frac{1 - \exp(-\eta_0^c)}{\exp(-\eta_0^c)} \times (2\bar{\mu} - \bar{\mu}\bar{K}_0\eta_0 - \eta_0^{d-1}\bar{\rho}(\eta_0, 0)). \tag{106}$$

Then, we study the order of both sides of equation 106 in terms of $\eta_0$ in the regime where $0 < \eta_0 \leq \min\left(\log\left(1 + \frac{\alpha_1}{2\alpha_2}\right)^{\frac{1}{c}}, \frac{1}{\bar{K}_0}\right)$. Since $0 < c < 1$, and the LHS of equation 106 is of order $\Theta(\eta_0^{1-c})$, it decreases monotonically to zero as $\eta_0$ approaches zero. The RHS of equation 106 is the product of two terms, i.e., $\frac{1 - \exp(-\eta_0^c)}{\exp(-\eta_0^c)}$ and $2\bar{\mu} - \bar{\mu}\bar{K}_0\eta_0 - \eta_0^{d-1}\bar{\rho}(\eta_0, 0)$. We notice $\eta_0^c$ approaches zero as $\eta_0$ decreases to zero. Thus, $\frac{1 - \exp(-\eta_0^c)}{\exp(-\eta_0^c)}$ converges to one as $\eta_0$ decreases to zero. Moreover, when $\eta_0 \leq \min\left(\log\left(1 + \frac{\alpha_1}{2\alpha_2}\right)^{\frac{1}{c}}, \frac{1}{\bar{K}_0}\right)$, we have $\bar{\mu} > 0$ and $0 < \bar{\rho}(\eta_0, 0) < 1$. Therefore, the RHS of equation 106 is of order $\Theta(1)$. As a result, when $\eta_0 > 0$ is sufficiently small, one has

$$4KL(0)\eta_0^2 < (1 - \exp(-\eta_0^c)) \times (1 - \Delta). \tag{107}$$

Moreover, if equation 102 has positive roots, and we use $\eta_0^{(2)}$ to denote its smallest positive root. The following holds for all $0 < \eta_0 \leq \eta_0^{(2)}$

$$4KL(0)\eta_0^2 \leq (1 - \exp(-\eta_0^c)) \times (1 - \Delta) \tag{108}$$

If equation 102 does not have positive root, then equation 107 holds for all positive $\eta_0$.

To summarize, we have shown that $\eta_{\max} > 0$, and the $\eta_0$ always exists. Moreover, when $\eta_0$ satisfies $0 < \eta_0 < \eta_{\max}$, the following holds

$$\bar{\mu} > 0, \quad 0 < \bar{\rho}(\eta_0, 0), \Delta < 1,$$
$$4KL(0)\eta_0^2 \leq (1 - \exp(-\eta_0^c)) \times (1 - \Delta). \tag{109}$$

Now we present the proof of Theorem E.1.

*Proof.* We employ an induction-based approach to prove Theorem E.1 by iteratively showing the following properties hold for all iteration $t$ when $\eta_0$ and $\eta_t$ satisfy the constraints in Theorem E.1.

- $A_1(t): L(t) \leq L(t-1)\rho(\eta_{t-1}, t-1) \leq L(t-1)\bar{\rho}(\eta_0, 0)$.

- $A_2(t): \beta_1 \leq \sigma_{\min}(W(t)) \leq \sigma_{\max}(W(t)) \leq \beta_2$.

- $A_3(t): \|D(t) - D(0)\|_F \leq \frac{2K\eta_0^2\alpha_2(0)\exp(\eta_0^c)L(0)}{1-\Delta}$.

- $A_4(t): \alpha_1 + 2\alpha_2\big(1 - \exp(\eta_0^c)\big) \leq \sigma_{\min}^2(\mathcal{T}_t) \leq \sigma_{\max}^2(\mathcal{T}_t) \leq \alpha_2\exp(\eta_0^c)$.

Assume $A_1(k), A_2(k), A_3(k), A_4(k)$ hold at iteration $k = 1, 2, \cdots, t$, then we show they all hold for iteration $t+1$.

**Prove $A_1(t+1)$ hold.**

We first show that under the constraints in Theorem E.1 and the induction assumption, one can lower bound and upper bound $\mu_t$ and $K_t$ using $\bar{\mu}$ and $\bar{K}_t$ respectively, which is characterized by the following lemma.

**Lemma E.1.** *The following lower bound and upper bound on $\mu_t$ and $K_t$ hold respectively*

$$\bar{\mu} \leq \mu_t, \quad K_t \leq \bar{K}_t. \tag{110}$$

The proof of the above lemma can be found at the end of Appendix E.

In Theorem 3.1, we have shown that the local PL inequality and Descent lemma hold with local PL constant $\mu_t$ and local smoothness constant $K_t$

$$L(t+1) \leq L(t) - \big(\eta_t - \frac{K_t\eta_t^2}{2}\big)\|\nabla L(t)\|_F^2, \quad \frac{1}{2}\|\nabla L(t)\|_F^2 \geq \mu_t L(t). \tag{111}$$

Therefore, one has

$$
\begin{aligned}
L(t+1) &\leq L(t) - \big(\eta_t - \frac{K_t\eta_t^2}{2}\big)\|\nabla L(t)\|_F^2 \\
&\leq L(t) - 2\mu_t\big(\eta_t - \frac{K_t\eta_t^2}{2}\big)L(t) && \text{Under the constraints } 0 < \eta_t < \frac{1}{K_t} \\
&\leq L(t) - 2\bar{\mu}\big(\eta_t - \frac{K_t\eta_t^2}{2}\big)L(t) && \text{Lemma E.1} \\
&= (1 - 2\bar{\mu}\eta_t + \bar{\mu}K_t\eta_t^2)L(t) \\
&\leq (1 - 2\bar{\mu}\eta_t + \bar{\mu}\bar{K}_t\eta_t^2)L(t) := \bar{\rho}(\eta_t, t)L(t) && \text{Lemma E.1}.
\end{aligned} \tag{112}
$$

Finally, we show $\rho(\eta_t, t) \leq \bar{\rho}(\eta_0, 0)$.

$$
\begin{aligned}
\rho(\eta_t, t) &\leq 1 - 2\bar{\mu}\eta_t + \bar{\mu}K_t\eta_t^2 \\
&\leq 1 - 2\bar{\mu}\eta_0 + \bar{\mu}K_t\eta_0^2 && \text{Use } \eta_0 \leq \eta_t \leq \frac{1}{K_t} \\
&\leq 1 - 2\bar{\mu}\eta_0 + \bar{\mu}\bar{K}_0\eta_0^2 := \bar{\rho}(\eta_0, 0) && \text{Use } K_t \leq \bar{K}_0.
\end{aligned} \tag{113}
$$

Therefore, $A_1(t+1)$ holds.

**Prove $A_2(t+1)$ hold.**

Since we have shown $A_1(t+1)$ holds, one has $L(t+1) \leq L(0)$. Moreover, based on the assumption that $\ell(W)$ is $\mu$-strongly convex and $K$-smooth, one has the following inequality

$$\frac{\mu}{2}\|W(t+1) - W^*\|_F^2 \leq \ell(t+1) = L(t+1) \leq \frac{K}{2}\|W(t+1) - W^*\|_F^2. \tag{114}$$

Then we can show $\sigma_{\max}(W(t+1)) \leq \beta_2$ as follows

$$
\begin{aligned}
\sigma_{\max}(W(t+1)) &= \sigma_{\max}(W(t+1) - W^* + W^*) \\
&\leq \sigma_{\max}(W^*) + \|W(t+1) - W^*\|_2 && \text{Weyl's inequality} \\
&\leq \sigma_{\max}(W^*) + \|W(t+1) - W^*\|_F \\
&\leq \sigma_{\max}(W^*) + \sqrt{\frac{2}{\mu}L(t+1)} \\
&\leq \sigma_{\max}(W^*) + \sqrt{\frac{2}{\mu}L(0)}. && \text{Use } L(t+1) \leq L(0)
\end{aligned}
\tag{115}
$$

For $\beta_1 \leq \sigma_{\min}(W(t+1))$, same result has been derived in Min et al. (2023). We refer the readers to Appendix B in Min et al. (2023) for details.

**Prove $A_3(t+1)$ hold.**

We first present the following lemma that bounds $\|D(k+1) - D(k)\|_F$ for all $k$.

**Lemma E.2.** *One has the following upper bound on $\|D(k+1) - D(k)\|_F$*

$$
\|D(k+1) - D(k)\|_F \leq 2K\eta_k^2 \sigma_{\max}^2(\mathcal{T}_k)L(k).
\tag{116}
$$

The proof of the above lemma can be found at the end of this section.

Based on Lemma E.2, one can show that $A_3(t+1)$ holds

$$
\begin{aligned}
\|D(t+1) - D(0)\|_F &\leq \sum_{k=0}^{t} \|D(k+1) - D(k)\|_F \\
&\leq \sum_{k=0}^{t} 2K\eta_k^2 \sigma_{\max}^2(\mathcal{T}_k)L(k) && \text{Lemma E.2} \\
&\leq \sum_{k=0}^{t} 2K\eta_k^2 \sigma_{\max}^2(\mathcal{T}_k)L(0)\bar{\rho}(\eta_0,0)^k && \text{Use } A_1(k), \forall k = 1, \cdots, t \\
&\leq \sum_{k=0}^{t} 2K\eta_k^2 \alpha_2 \exp(\eta_0^c)L(0)\bar{\rho}(\eta_0,0)^k && \text{Use } A_4(k), \forall k = 1, \cdots, t \\
&\leq \sum_{k=0}^{t} 2K(1+\eta_0^d)^k \eta_0^2 \alpha_2 \exp(\eta_0^c)L(0)\bar{\rho}(\eta_0,0)^k && \text{Use } \eta_k \leq (1+\eta_0^d)^{\frac{k}{2}}\eta_0 \\
&= 2KL(0)\exp(\eta_0^c)\eta_0^2 \alpha_2 \sum_{k=0}^{t} \Delta^k && \Delta = (1+\eta_0^d)\bar{\rho}(\eta_0,0) \\
&\leq \frac{2K\eta_0^2 \alpha_2 \exp(\eta_0^c)L(0)}{1 - \Delta}. && 0 < \Delta < 1
\end{aligned}
\tag{117}
$$

**Prove $A_4(t+1)$ hold.**

We first present the following two lemmas which will be used to prove that $A_4(t+1)$ hold.

**Lemma E.3.** *One can use $\alpha_1, \alpha_2$ to lower and upper bound the singular values of $\mathcal{T}_0$*

$$
\alpha_1 \leq \sigma_{\min}^2(\mathcal{T}_0) \leq \sigma_{\max}^2(\mathcal{T}_0) \leq \alpha_2.
\tag{118}
$$

**Lemma E.4.** *One can bound the deviation of the singular values of $\mathcal{T}_k$ using the deviation of the imbalance $\|D(k) - D(0)\|_F$*

$$
\sigma_{\min}^2(\mathcal{T}_k) \geq \alpha_1 - 4\|D(k) - D(0)\|_F := \mathcal{T}_k^L.
\tag{119}
$$

$$
\sigma_{\max}^2(\mathcal{T}_k) \leq \alpha_2 + 2\|D(k) - D(0)\|_F := \mathcal{T}_k^U.
\tag{120}
$$

The proof of Lemma E.3 and Lemma E.4 can be in Xu et al. (2023), Appendix C.

Notice $\mathcal{T}_{t+1}^{\mathrm{L}} + 2\mathcal{T}_{t+1}^{\mathrm{U}} = \alpha_1 + 2\alpha_2$. Therefore, if one can show

$$\mathcal{T}_{t+1}^{\mathrm{U}} \leq \exp(\eta_0^c)\alpha_2. \tag{121}$$

Then, the following holds directly

$$\sigma_{\max}^2(\mathcal{T}_{t+1}) \leq \mathcal{T}_{t+1}^{\mathrm{U}} \leq \exp(\eta_0^c)\alpha_2, \tag{122}$$

$$\sigma_{\min}^2(\mathcal{T}_{t+1}) \geq \mathcal{T}_{t+1}^{\mathrm{L}} = \alpha_1 + 2\alpha_2 - 2\mathcal{T}_{t+1}^{\mathrm{U}} \geq \alpha_1 + 2\alpha_2\big(1 - \exp(\eta_0^c)\big). \tag{123}$$

Therefore, it suffices to show equation 121 holds. We start from Lemma E.4

$$
\begin{aligned}
\mathcal{T}_k^{\mathrm{U}} &= \alpha_2 + 2\|D(k) - D(0)\|_F \\
&\leq \alpha_2 + \frac{4KL(0)\eta_0^2\alpha_2(0)\exp(\eta_0^c)}{1 - \Delta} && \text{Use } A_3(t+1) \\
&\leq \alpha_2 + (1 - \exp(-\eta_0^c)) \times (1 - \Delta) \cdot \frac{\alpha_2\exp(\eta_0^c)}{1 - \Delta} && \text{Equation 107} \\
&= \exp(\eta_0^c)\alpha_2.
\end{aligned}
\tag{124}
$$

$\square$

Now, we present the proof of lemmas used in the proof of Theorem E.1. All lemmas presented below are based on the assumption that $A_1(k), A_2(k), A_3(k), A_4(k)$ hold for all iterations $k = 1, 2, \cdots, t$ and the constraints presented in Theorem E.1. For convenience, we do not state these assumptions and constraints repetitively.

**Lemma E.1.** *The following lower bound and upper bound on $\mu_t$ and $K_t$ hold respectively*

$$\bar{\mu} \leq \mu_t, \quad K_t \leq \bar{K}_t. \tag{125}$$

*Proof.* We start with the lower bound on $\mu_t$. Due to the assumption that $A_4(t)$ hold, one has the following lower bound $\mu_t$

$$\mu_t = \mu\sigma_{\min}^2(\mathcal{T}_t) \geq \mu\alpha_1. \tag{126}$$

For the upper bound on $K_t$, we first show that based on the assumption that $A_1(k)$ hold for all $k \leq t$, one has

$$L(t) \leq L(t-1)\bar{\rho}(\eta_0, 0) \leq L(0)\bar{\rho}(\eta_0, 0)^t. \tag{127}$$

Then, based on equation 127, $A_4(t)$ and the constraint that $\eta_t \leq (1 + \eta_0^d)^{\frac{t}{2}}\eta_0$, we can derive the following upper bound on $K_t$

$$
\begin{aligned}
K_t &= K\sigma_{\max}^2(\mathcal{T}_t) + \sqrt{2KL(t)} + 6K^2\sigma_{\max}(W(t))L(t)\eta_t^2 + 3K\sigma_{\max}^2(\mathcal{T}_t)\sqrt{2KL(t)}\eta_t \\
&\leq K\alpha_2\exp(\eta_0^c) + \sqrt{2KL(0)\bar{\rho}(\eta_0, 0)^t} + 6K^2\beta_2 L(0)\bar{\rho}(\eta_0, 0)^t\eta_t^2 \\
&\quad + 3K\alpha_2\exp(\eta_0^c)\sqrt{2KL(0)\bar{\rho}(\eta_0, 0)^t}\eta_t \\
&\leq K\alpha_2\exp(\eta_0^c) + \sqrt{2KL(0)\bar{\rho}(\eta_0, 0)^t} + 6K^2\beta_2 L(0)\bar{\rho}(\eta_0, 0)^t(1 + \eta_0^d)^t\eta_0^2 \\
&\quad + 3K\alpha_2\exp(\eta_0^c)\sqrt{2KL(0)\bar{\rho}(\eta_0, 0)^t}(1 + \eta_0^d)^{\frac{t}{2}}\eta_0 && \text{Use } \eta_t \leq (1 + \eta_0^d)^{\frac{t}{2}}\eta_0 \\
&= \sqrt{2KL(0)\bar{\rho}(\eta_0, 0)^t} + 6K^2\beta_2 L(0)\eta_0^2\Delta^t + K\exp(\sqrt{\eta_0})\alpha_2\big[1 + 3\sqrt{2KL(0)\Delta^t}\eta_0\big],
\end{aligned}
\tag{128}
$$

where the last line follows the definition of $\Delta = (1 + \eta_0^d)\bar{\rho}(\eta_0, 0)$. $\square$

**Lemma E.2.** *One has the following upper bound on $\|D(k+1) - D(k)\|_F$*

$$\|D(k+1) - D(k)\|_F \leq 2K\eta_k^2\sigma_{\max}^2(\mathcal{T}_k)L(k). \tag{129}$$

*Proof.* In equation 7 and equation 8, we have

$$W_1(k+1) = W_1(k) - \eta_k \nabla \ell(k) W_2(k), \quad W_2(k+1) = W_2(k) - \eta_k \nabla \ell(k)^\top W_1(k). \tag{130}$$

There, we can compute $D(k+1) - D(k)$ as follows

$$
\begin{aligned}
D(k+1) - D(k) =& W_1(k+1)^\top W_1(k+1) - W_2(k+1)^\top W_2(k+1) \\
& - W_1(k)^\top W_1(k) + W_2(k)^\top W_2(k) \\
=& \left(W_1(k) - \eta_k \nabla \ell(k) W_2(k)\right)^\top \left(W_1(k) - \eta_k \nabla \ell(k) W_2(k)\right) \\
& - \left(W_2(k) - \eta_k \nabla \ell(k)^\top W_1(k)\right)^\top \left(W_2(k) - \eta_k \nabla \ell(k)^\top W_1(k)\right) \\
& - W_1(k)^\top W_1(k) + W_2(k)^\top W_2(k) \\
=& \eta_k^2 \left(W_2(k)^\top \nabla \ell(k)^\top \nabla \ell(k) W_2(k) - W_1(k)^\top \nabla \ell(k)^\top \nabla \ell(k) W_1(k)\right). 
\end{aligned}
\tag{131}
$$

Based on the above equation, one can bound $\|D(k+1) - D(k)\|_F$ as follows

$$
\begin{aligned}
\|D(k+1) - D(k)\|_F =& \eta_k^2 \|W_2(k)^\top \nabla \ell(k)^\top \nabla \ell(k) W_2(k) - W_1(k)^\top \nabla \ell(k)^\top \nabla \ell(k) W_1(k)\|_F \\
\text{Property of norm } \leq& \eta_k^2 \|W_2(k)^\top \nabla \ell(k)^\top \nabla \ell(k) W_2(k)\|_F + \eta_k^2 \|W_1(k)^\top \nabla \ell(k)^\top \nabla \ell(k) W_1(k)\|_F \\
\text{equation 37 } \leq& \eta_k^2 \sigma_{\max}^2(W_2(k)) \|\nabla \ell(k)\|_F^2 + \eta_k^2 \sigma_{\max}^2(W_1(k)) \|\nabla \ell(k)\|_F^2 \\
=& \eta_k^2 \sigma_{\max}^2(\mathcal{T}_k) \|\nabla \ell(k)\|_F^2 \\
K\text{-smooth of } \ell \leq& 2K \eta_k^2 \sigma_{\max}^2(\mathcal{T}_k) L(k). 
\end{aligned}
\tag{132}
$$

$\square$

# F    Verification of the assumption $\alpha_1 > 0$

In this section, we provide two conditions that ensure $\alpha_1 > 0$.

In Min et al. (2021), the authors show the following lemma which guarantees $\alpha_1 > 0$.

**Lemma F.1** (Lemma 1 in (Min et al., 2021)). *Let $W_1(0), W_2(0)$ are initialized entry-wise i.i.d. from $\mathcal{N}(0, \frac{1}{h^{2p}})$ with $\frac{1}{4} \leq p \leq \frac{1}{2}$. For $\forall \delta > 0$ and $h \geq poly(n, m, \frac{1}{\delta})$, with probability $1 - \delta$ over random initialization with $W_1(0), W_2(0)$, the following holds*

$$\alpha_1 \geq h^{1-2p}. \tag{133}$$

The above theorem states when Problem 2 is **sufficiently overparametrized**, i.e., $h \geq \text{poly}(n, m, \frac{1}{\delta})$, Gaussian initialization with proper variance ensures $\alpha_1$ has a positive lower bound $h^{1-2p}$. Moreover, the lower bound increases as $h$ increases.

Next, we are going to show with **mild** overparametrization, one can ensure $\alpha_1 > 0$.

**Lemma F.2** (Mild overparametrization ensures $\alpha_1 > 0$). *Let $W_1(0), W_2(0)$ are initialized entry-wise i.i.d. from a continuous distribution $\mathbb{P}$. When $h \geq m+n$, the following holds almost surely over random initialization with $W_1(0), W_2(0)$*

$$\alpha_1 > 0. \tag{134}$$

Compared with Lemma F.1, Lemma F.2 considers a wider range of distributions that include Gaussian distribution and uniform distribution. Thus, commonly used random initialization schemes, such as Xavier initialization (Glorot & Bengio, 2010) and He initialization (He et al., 2015), lead to $\alpha_1 > 0$. Moreover, the requirement of overparametrization in Lemma F.2 is mild compared with the one in Lemma F.1, i.e., $h \geq m+n$ versus $h \geq \text{poly}(n, m, \frac{1}{\delta})$. As a result, Lemma F.2 can be applied to more general overparametrization. On the other hand, the conclusion of Lemma F.2 is weaker than Lemma F.1 in the sense that Lemma F.2 only proves $\alpha_1 > 0$ but do not characterize its magnitude.

Before presenting the proof of Lemma F.2, we first present two lemmas that will be used in the proof.

**Lemma F.3.** *Let $A \in \mathbb{R}^{h \times n}, h \geq n$ be a random matrix with entry-wise drawn i.i.d. from a continuous distribution $\mathbb{P}$. Then $A$ is of full column rank almost surely.*

We refer the readers to (Vershynin, 2018) for detailed proof.

**Lemma F.4.** *A sufficient condition for $\alpha_1 > 0$ is $\sigma_{m+n}(D(0)) > 0$.*

The proof of this lemma can be found in (Min et al., 2021).

Now we present the proof of Lemma F.2

*Proof.* Based on Lemma F.4, it suffices to show that one almost surely has $\sigma_{m+n}(D(0)) > 0$ over random initialization with $W_1(0), W_2(0)$. We use proof by contradiction. Assume $\sigma_{m+n}(D(0)) = 0$, then one has $\dim(\ker D(0)) \geq h - n - m + 1$.

On the other hand, Lemma F.3 implies with probability one, $[W_1^\top(0), W_2^\top(0)] \in \mathbb{R}^{h \times (n+m)}$ is of full column rank. Our next step is to show $\dim(\ker D(0)) \leq h - n - m$. If this is true, then there is a contradiction. Thus, one directly has $\sigma_{m+n}(D(0)) > 0$.

For any $v \in \mathbb{R}^h$ that satisfies $D(0)v = 0$, we can write this equation as follows

$$D(0)v = 0 \Leftrightarrow [W_1^\top(0), W_2^\top(0)] \begin{bmatrix} W_1(0) \\ -W_2(0) \end{bmatrix} v = 0 \tag{135}$$

Since $[W_1^\top(0), W_2^\top(0)]$ is of full column rank, the above equation is equivalent to

$$\begin{bmatrix} W_1(0) \\ -W_2(0) \end{bmatrix} v = 0, \tag{136}$$

and $\dim(\ker D(0)) \leq h - n - m$. $\qquad\square$

## F.1 Large width and proper choices of the variance lead to well-conditioned $\mathcal{T}_0$

In this section, we provide proof for Theorem 3.3. We first restate Theorem 3.3 here.

**Theorem F.1** (Restate of Theorem 3.3)**.** *Let $W_1(0), W_2(0)$ are initialized entry-wise i.i.d. from $\mathcal{N}(0, \frac{1}{h^{2p}})$ with $\frac{1}{4} < p < \frac{1}{2}$. . $\forall \delta \in (0, 1), \forall h \geq poly(m, n, \frac{1}{\delta})$, with probability $1 - \delta$ over random initialization $W_1(0), W_2(0)$, the following condition holds*

$$\frac{\sigma_{\min}(\mathcal{T}_0)}{\sigma_{\max}(\mathcal{T}_0)} \geq \frac{\alpha_1}{\alpha_2} \geq 1 - \Omega(h^{2p-1}). \tag{137}$$

*Proof.* Since $\alpha_1, \alpha_2$ are the lower and upper bounds for the singular values of $\mathcal{T}_0$, it is straightforward to see

$$\frac{\sigma_{\min}(\mathcal{T}_0)}{\sigma_{\max}(\mathcal{T}_0)} \geq \frac{\alpha_1}{\alpha_2}. \tag{138}$$

Thus, it suffices to show $\frac{\alpha_1}{\alpha_2} \geq 1 - \Omega(h^{p-\frac{1}{2}})$. We provide upper bounds and lower bounds of $\alpha_1, \alpha_2$ separately to prove Theorem 3.3. In Min et al. (2022), the authors provide the following lower bound on $\alpha_1$ under the same setting as Theorem 3.3.

**Lemma F.5** (Lemma 11 in Min et al. (2022))**.** *Under the same setting as Theorem 3.3, with probability over $1 - \delta$ over random initialization over $W_1(0), W_2(0)$, the following holds*

$$\alpha_1 \geq 2h^{1-2p} + 2B^2 h^{-2p} - 4Bh^{\frac{1}{2}-2p},$$
$$\left| W_1(0)W_2^\top(0) \right|_F \leq 2\sqrt{m}Bh^{\frac{1}{2}-2p}, \tag{139}$$

*where $B = \sqrt{m+n} + \frac{1}{2}\log\frac{2}{\delta}$.*

We refer the readers to Min et al. (2022) for detailed proof.

Then, we derive an upper bound on $\alpha_2$ where the definition is given as (See also Table 2)

$$\alpha_2 = \frac{\lambda_+ + \sqrt{\lambda_+^2 + 4\beta_2^2}}{2} + \frac{\lambda_- + \sqrt{\lambda_-^2 + 4\beta_2^2}}{2} , \tag{140}$$

where $\lambda_-, \lambda_+, \beta_2$ is defined as follows

$$\lambda_- = \max(\lambda_{\max}(-D(0)), 0), \quad \lambda_+ = \max(\lambda_{\max}(D(0)), 0) ,$$

$$\beta_2 = \sigma_{\max}(W^*) + \sqrt{\frac{K}{\mu}} \|W_1(0)W_2^\top(0) - W^*\|_F . \tag{141}$$

Based on Lemma F.5, one can upper bound $\beta_2$ as follows

$$\beta_2 = \sigma_{\max}(W^*) + \sqrt{\frac{K}{\mu}} \|W_1(0)W_2^\top(0) - W^*\|_F$$

$$\leq \sigma_{\max}(W^*) + \sqrt{\frac{K}{\mu}} \left(\|W^*\|_F + 2\sqrt{m}Bh^{\frac{1}{2}-2p}\right) , \tag{142}$$

Then, we provide an upper bound for $\lambda_{\max}(D(0))$. Similar analysis can be used to derive an upper bound for $\lambda_{\max}(-D(0))$.

First, based on Lemma A.3, the following holds with probability at least $1 - \delta$

$$\sigma_{\max}(h^p[W_1(0), W_2(0)]) \leq \sqrt{h} + \sqrt{m+n} + \frac{1}{2}\log\frac{2}{\delta} . \tag{143}$$

Therefore, $\sigma_{\max}([W_1, W_2]) \leq h^{\frac{1}{2}-p} + Bh^{-p}$. Then, based on this upper bound, one can derive the following upper bound on $\sigma_{\max}(D(0))$

$$\sigma_{\max}(D(0)) = \sigma_{\max}\left( \left[W_1(0), W_2(0)\right] \begin{bmatrix} W_1^\top(0) \\ -W_2^\top(0) \end{bmatrix} \right)$$

$$\leq \sigma_{\max}\left( \left[W_1(0), W_2(0)\right] \right) \times \sigma_{\max}\left( \begin{bmatrix} W_1^\top(0) \\ -W_2^\top(0) \end{bmatrix} \right)$$

$$= \sigma_{\max}^2\left( \left[W_1(0), W_2(0)\right] \right)$$

$$\leq h^{1-2p} + 2h^{\frac{1}{2}-2p}B + B^2h^{-2p} . \tag{144}$$

Similarly, one can show $\sigma_{\max}(-D(0)) \leq h^{1-2p} + 2h^{\frac{1}{2}-2p}B + B^2h^{-2p}$. By combining all the results, one can show the following upper bound on $\alpha_2$

$$\alpha_2 = \frac{\lambda_+ + \sqrt{\lambda_+^2 + 4\beta_2^2}}{2} + \frac{\lambda_- + \sqrt{\lambda_-^2 + 4\beta_2^2}}{2}$$

$$\leq \lambda_+ + \lambda_- + 2\beta_2$$

$$\leq \sigma_{\max}(D(0)) + \sigma_{\max}(-D(0)) + 2\beta_2$$

$$\leq 2h^{1-2p} + 4h^{\frac{1}{2}-2p}B + 2B^2h^{-2p} + 2\sigma_{\max}(W^*) + 2\sqrt{\frac{K}{\mu}}\left(\|W^*\|_F + 2\sqrt{m}Bh^{\frac{1}{2}-2p}\right) .$$

With the bounds on $\alpha_1, \alpha$, one can derive the following bound on $\frac{\sigma_{\min}(\mathcal{T}_0)}{\sigma_{\max}(\mathcal{T}_0)}$

$$
\begin{aligned}
\frac{\sigma_{\min}(\mathcal{T}_0)}{\sigma_{\max}(\mathcal{T}_0)} &\geq \frac{\alpha_1}{\alpha_2} \\
&\geq \frac{2h^{1-2p} + 2B^2 h^{-2p} - 4Bh^{\frac{1}{2}-2p}}{2h^{1-2p} + 4h^{\frac{1}{2}-2p}B + 2B^2 h^{-2p} + 2\sigma_{\max}(W^*) + 2\sqrt{\frac{K}{\mu}}\left(\|W^*\|_F + 2\sqrt{m}Bh^{\frac{1}{2}-2p}\right)} \\
&= 1 - \frac{8h^{\frac{1}{2}-2p}B + 2\sigma_{\max}(W^*) + 2\sqrt{\frac{K}{\mu}}\left(\|W^*\|_F + 2\sqrt{m}Bh^{\frac{1}{2}-2p}\right)}{2h^{1-2p} + 4h^{\frac{1}{2}-2p}B + 2B^2 h^{-2p} + 2\sigma_{\max}(W^*) + 2\sqrt{\frac{K}{\mu}}\left(\|W^*\|_F + 2\sqrt{m}Bh^{\frac{1}{2}-2p}\right)}, \\
&= 1 - \Omega(h^{2p-1}),
\end{aligned}
\tag{145}
$$

where the last line holds because the dominating term in the numerator and denominator is of order $\mathcal{O}(1)$ and $\mathcal{O}(h^{1-2p})$ separately. $\qquad\square$

## G   Simulation

In this section, we first numerically verify that with proper initialization (See Theorem 3.3), a larger width will lead to a well-conditioned $\mathcal{T}_0$. Then, we present experiments showing that the overparametrized model trained with GD following the adaptive step size proposed in §3.2 can almost match the rate of non-overparametrized model. Throughout the experiments, we consider  Problem 2 with squared loss

$$
L(W_1, W_2) = \frac{1}{2}\|Y - XW_1 W_2^\top\|_F^2.
\tag{146}
$$

### G.1   Large width leads to well-conditioned $\mathcal{T}_0$

In this section, we compare the $\kappa(\mathcal{T}_0)$ under different scales of the variance of the initialization, i.e., $p$, and difference width of the networks, i.e., $h$. We choose $p \in \{0.275, 0.375, 0.475\}$ and $h$ from $[500, 2000]$. We generate the data matrix $X$ as a $10 \times 10$ orthogonal matrix and $Y = X\Theta + \mathcal{N}(0, 0.1)$ where $\Theta \in \mathbb{R}^{10 \times 10}$ is entry-wise i.i.d. sampled from $\mathcal{N}(0, 0.1)$. The weight matrices $W_1, W_2$ are initialized following Theorem 3.3.

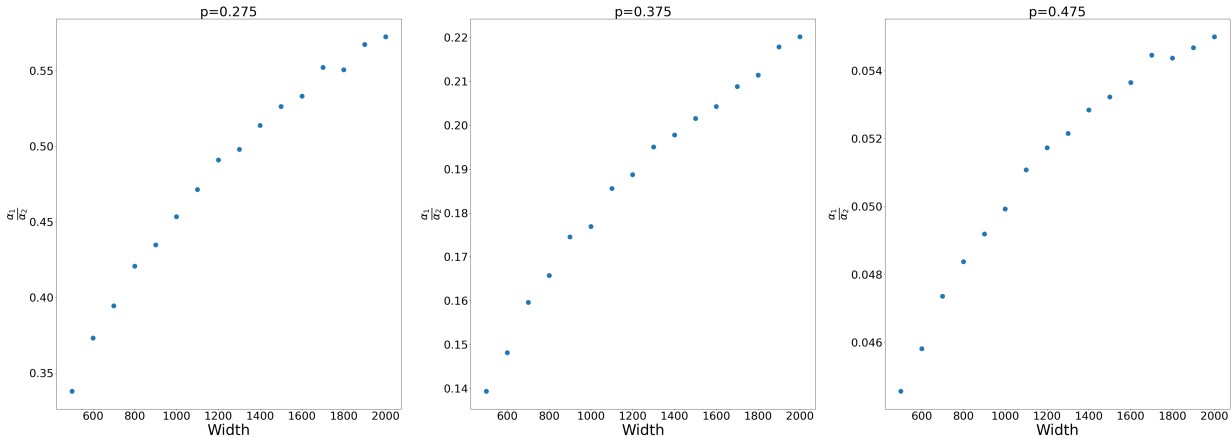

Figure 4: $\kappa(\mathcal{T}_0)$ under different choices of $p$ and $h$. We repeat the simulation thirty times and plot the average value of $\kappa(\mathcal{T}_0)$.

Figure4 shows that for fixed $p$, larger width leads to well-conditioned $\kappa(\mathcal{T}_0)$. Moreover, for a fixed width, smaller $p$ will lead to smaller $\frac{\alpha_1}{\alpha_2}$. In Theorem 3.3, we show $\frac{\alpha_1}{\alpha_2} \geq 1 - \Omega(h^{p-\frac{1}{2}})$. One can see if we decrease either $h$ or $p$, the lower bound on $\frac{\alpha_1}{\alpha_2}$ decreases. Therefore, the simulation results support Theorem 3.3.

## G.2 Overparametrized GD can almost match the rate of the non-overparametrized GD

In this section, we compare the rate of overparametrized GD following the adaptive step size in §3.2 with the non-overparametrized GD. We initialize weight matrices $W_1, W_2 \in \mathbb{R}^{5 \times h}$ as entry-wise $\mathcal{N}(0, \frac{1}{h})$. The data matrices are generated as $X = U\Sigma V, Y = XW_1W_2^\top + \mathcal{N}(0, 0.1)$ where $U, V \in \mathbb{R}^{5 \times 5}$ are random orthogonal matrices and $\Sigma \in \mathbb{R}^{5 \times 5}$ is a diagonal matrix where the diagonal entry is uniformly i.i.d. drawn from $[1.8, 2.3]$. The step size for the non-overparametrized GD is $\eta_t = \frac{1}{\sigma_{\max}^2(X)}$, and the step size for the overparametrized GD follows equation 19 with $h(\eta_t, t) = \rho(\eta_t, t)$. Figure 5 shows that as one increases the width of the networks, the

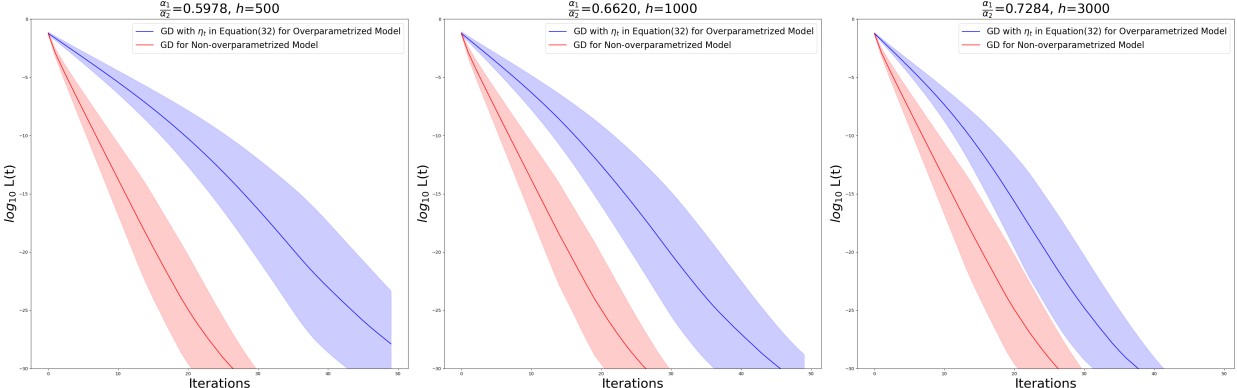

Figure 5: Comparison of convergence rate of GD for the non-overparametrized model and overparametrized model. We run the simulations thrity times. The red line represents $\log_{10} \ell(t)$ and the blue line represents $\log_{10} L(t)$. The shaded area represents plus and minus one standard deviation of the reported loss.

overparametrized GD can almost match the rate of the non-overparametrized GD asymptotically. Moreover, as the width increases, $\frac{\alpha_1}{\alpha_2}$ increases and the rate of the overparametrized GD is more close to the one of the non-overparametrized GD. This is because, in §3, we show that the optimal local rate of convergence can be arbitrarily close to $1 - \frac{\mu}{K} \cdot \frac{\alpha_1}{\alpha_2}$. Therefore, as one increases the width, $\frac{\alpha_1}{\alpha_2}$ approaches one, and this leads to the rate of overparametrized GD approaches $1 - \frac{\mu}{K}$.

### G.2.1 Detailed description of backtracking line search

For backtracking line search, the algorithm is described as follows: In the simulation in §4.2, we choose $\tau = 0.1$

---

**Algorithm 1** Backtracking Line Search.

---
**Given** Data matrices $X, Y$, initialization $W_1(0), W_2(0)$, and hyperparameters $\eta_{bt}, \tau, \gamma$.
**Result** $W_1^*, W_2^*$ that minimize $L(W_1, W_2) = \frac{1}{2}\|Y - XW_1W_2^\top\|_F^2$.
**for** $t = 0, 1, \cdots, T$ **do**
  $\eta_t = \eta_{bt}$
  **while** $L(W_1(t) - \eta_t \nabla_{W_1} L(t), W_2(t) - \eta_t \nabla_{W_2} L(t)) > L(t) - \gamma\|\nabla L(t)\|_F^2$ **do**
    $\eta_t = \tau \eta_t$
  **end while**
  $W_1(t+1) = W_1(t) - \eta_t \nabla_{W_1} L(t)$
  $W_2(t+1) = W_2(t) - \eta_t \nabla_{W_2} L(t)$
**end for**

---

and $\gamma = 0.9$.

Figure4 shows that for fixed $p$, larger width leads to well-conditioned $\kappa(\mathcal{T}_0)$. Moreover, for a fixed width, smaller $p$ will lead to smaller $\frac{\alpha_1}{\alpha_2}$. In Theorem 3.3, we show $\frac{\alpha_1}{\alpha_2} \geq 1 - \Omega(h^{p-\frac{1}{2}})$. One can see if we decrease either $h$ or $p$, the lower bound on $\frac{\alpha_1}{\alpha_2}$ decreases. Therefore, the simulation results support Theorem 3.3.

