# OpenReview forum: "A Local Polyak-Łojasiewicz and Descent Lemma of Gradient Descent For Overparametrized Linear Models"
_TMLR — Accepted by TMLR_

### Review · Reviewer_QAmF · 2025-03-13

**Summary Of Contributions:**

This paper studies the convergence of overparamterized linear networks for GD. As the classical ingredients such as PL condition and smoothness condition do not hold globallay for these losses, the authors propose a local analysis. They show that instead it is possible to propose a local PL and smoothness condition that depends on the condition number of the layer factors. This quantity may change say for example to near a saddle or a global minima, still allowing them to prove convergence.

**Audience:**

Yes

**Claims And Evidence:**

Yes

**Requested Changes:**

Please see the points mentioned in the weakness. Addressing 1-4 and citing the papers in proper context would be beneficial.

**Strengths And Weaknesses:**

This is a nice paper with stengths:

1) The problem is important since convergence analysis for overparamterzied networks are important.
2) The experiments and theory matches well. The difficulty at hand is clearly explained.
3) Although their exist past works that uses local PL condition to prove convergence in the same setting. The **criteria for TMLR is not novelty** but that claims should support evidence. The later condition holds for this paper, but not necessarily the first one.

Here are some weakness that can be addresable:


1) Please discuss how the local PL inequality behaves near a saddle. seems like the coefficient needs to be very small. some landscape diagram can help reader grasp the problem. for example 1d comparisons of (x-1)^2 and (x^2-1)^2 and showing that PL condition is violated near the saddle of the latter, can be a good way to show this.

2) local pseudo convexity (instead of strong convexity) have been previously used in problems such as in https://arxiv.org/pdf/2310.03956. Better to cite such papers where these local PL conditions are previously used in analysis.

3) Some comparisons to work such as https://arxiv.org/pdf/2409.19791 are missing, which also shows linear convergence on adaptive step size GD on such overparamterized non-convex optimization problems. However, the choice of adaptive steps are different from the current work. But a discussion should be helpful.

4) What if a constant step-size is used? then what is the largest step-size that can be used to ensure convergence? Does this step-size depend on the initialization or imbalance? Works such as https://openreview.net/forum?id=J4Dvxv7WnG show that the step-size that can drive overparamterized linear networks is independent of such initlaization or imbalance. More specifically for GD, the imbalance gap monotonically decays with linear convergence rate even beyond the stability threshold. Moreover, even beyond this stability limit, the loss do not diverge but shows stable oscillations. Discussing the observations are helpful since the current work studies the descent lemma.

---

### Review · Reviewer_gdTu · 2025-03-16

**Summary Of Contributions:**

The paper studies the convergence of the gradient descent in linear networks and derives a linear local convergence rate.
To achieve this, the submission proves the local versions of the Polyak-Łojasiewicz condition and Descent lemma in Theorem 3.1 and, based on them, obtains the local convergence results in Theorem 3.2.
Similarly to recent related works, the preprint allows for finitely wide networks and common initialization schemes, and it
improves on other results by allowing for more general loss functions and adaptive learning rates. The theoretical bounds on the convergence rates are verified experimentally.
Finally, the work derives an algorithm for the step-size choice based on Theorem 3.2 and demonstrates experimental improvements over the previous methods.

**Audience:**

Yes

**Broader Impact Concerns:**

None.

**Claims And Evidence:**

Yes

**Requested Changes:**

**Crucial**

C1. Add a discussion on which loss functions would satisfy the assumptions.
Where possible, rephrase the general loss claim. General loss sounds like any or almost any loss is acceptable.

C2. Enhancing the clarity of the paper:

1. Reference the specific statements in the Main Contribution section.

2. Reference assumptions in the theorem statements: assumptions on the loss and other assumptions used (if any). The assumption that $h \geq \min (n, m)$ is also required to obtain the results, as I understand.

3. Add names for the quantities where possible in Table 2.

4. Add proof intuitions for the theorems or mention the key techniques used.

**Minor**

C1. Specify what an overparametrized network means. Definitions of Problem 1 and Problem 2, as well as definitions of the overparametrized and non-overparametrized problems at the top of page 2, suggest that overparametrization is a property that depends on the network depth.

C2. Specify what LS is on page 2.

**Strengths And Weaknesses:**

**Strengths**

S1. Given the complexity of the global convergence analysis of neural networks, as is also discussed in the paper, analysis of the local convergence in two-layer linear networks naturally can help further the understanding of the problem. To my knowledge, such results have not appeared before in the setup introduced in the paper.

S2. The paper uses clear language, and there are in-detail explanations for a reasonable number of questions a reader might have. For instance, there are explanations for why studying global convergence is problematic, when the initialization requirements are satisfied, and why the step size requirements are defined a certain way.

**Weaknesses**

W1. There is no discussion on which commonly used losses would satisfy the loss assumptions.

W2. While I appreciated the thorough explanations, all the small details, sometimes also distributed across the paper, made it overwhelming and hard to follow at times. I understand that, to some extent, it is because of the nature of the results. Nevertheless, I suggest what can be improved in the Requested Changes.

---

### Review · Reviewer_wrn2 · 2025-03-16

**Summary Of Contributions:**

The paper investigates the convergence of GD for training two-layer linear neural networks, addressing limitations in prior work that relied on stringent assumptions about step size, network width, and initialization. By leveraging the PL condition and the Descent Lemma, tthe authors bound constants dependent on initialization and loss, proving linear convergence for general losses. They propose an adaptive step size scheme, validated empirically, which accelerates convergence compared to fixed step sizes. The analysis extends to matrix factorization, improving singular value tracking for adaptive step size computation.

**Audience:**

Yes

**Broader Impact Concerns:**

I have not found any discussions about the limitations and potential negative societal impact. But in my opinion, this may not be a problem, since the work only focuses on the analyzing the convergence of GD. Still, it is highly encouraged to add corresponding discussions.

**Claims And Evidence:**

Yes

**Requested Changes:**

See weakness.

**Strengths And Weaknesses:**

**Strengths**

1. The paper is clearly written and easy to follow. Also, the presentation of the paper is good.
2. The topic of this paper seems interesting, investigating the the convergence of GD for training two-layer linear neural networks is the base for understanding the convergence behavior for training deep neural networks.
3. The authors provide rigorous theoretical analysis, analyzing linear convergence across diverse scenarios as well as offering a comprehensive analysis of how factors like step size, network width, and initialization influence the process, which to my knowledge could be an interesting results to the field.
4. The authors provide some valuable empirical results.

** Weakness **

In my view, the primary concerns relate to the significance of the contributions in comparison to existing literature. Specifically, when compared to closely related works [1, 2], which also establish linear convergence, particularly [2], which considers a very similar setting except for the step-size constraint, I could not say the contribution is quite significant. A more substantial contribution might involve extending the analysis to a different dimension. At present, a more meaningful results would be to demonstrate the convergence analysis in the context of deep networks or under the suboptimal assumption.

[1]  Arora, Sanjeev, et al. "A convergence analysis of gradient descent for deep linear neural networks." arXiv preprint arXiv:1810.02281 (2018).

[2] Xu, Ziqing, et al. "Linear convergence of gradient descent for finite width over-parametrized linear networks with general initialization." International Conference on Artificial Intelligence and Statistics. PMLR, 2023.

Typos, Page 5, "such an analysis to Problem Problem 2. " -> "such an analysis to Problem 2. "

---

### Decision · Action_Editor_yE7U · 2025-04-18

**Recommendation:** Accept with minor revision

**Comment:**

The authors have studied convergence of gradient descent (GD) for two-layer linear networks under a general loss and finite step-size. The key contribution is to show that a local PL inequality and a local smoothness inequality hold along the trajectory of GD leading to local descent lemma for such networks. The authors further leverage on this descent lemma to design  an adaptive step-size, which achieves a linear convergence rate for GD.

The reviewers have initially raised concerns regarding comparison with related work that use local PL inequality to achieve convergence and those obtaining linear convergence rate for linear networks under different step-sizes compared to this paper. The authors have clarified that their analysis explicitly characterizes how the network width and the initialization scheme impact the convergence rate. The reviewers acknowledge rigorous theoretical results with providing sufficient insights and discussion on the key theorems as major strengths of this paper.


In the revised version, the authors have clarified their notion of overparameterization on page 2 “Here, overparametrized indicates that, although the function classes represented by $XW$ and $XW_1W_2\cdots W_L$ coincide under certain width assumptions, we introduce additional matrices to represent the same function. The overparametrized loss above can be seen as using a deep linear network to learn the target $Y$.” So depth of the network is the key point. On page 4 and the rest of the paper, they introduce (Problem 2) for two-layer networks as an overparameterized problem. This is not consistent with the notion earlier discussed on page 2. I’d recommend that the paper is accepted after addressing this comment.

**Audience:**

Yes

**Claims And Evidence:**

Yes

---

> ### Author Response · Authors · 2025-05-16
> **Revision in Camera-ready version**
>
> Dear Action Editor,
>
> Thank you for your efforts in reviewing our submissions and valuable suggestions. We have revised the paper accordingly. In particular, per your recommendation, we have updated page 2 where we introduce the notion of overparameterization:
>
> We now introduce the general case for network depth $L>2$ to align with existing work and to provide full context. We then clarify that, in this paper, we focus specifically on the $L=2$ setting.
>
> We hope these changes address your concerns. Please let us know if any further modifications are needed.
>
> Thank you again for your time and guidance.